EMBO
Molecular Medicine

# 4′-Phosphopantetheine corrects CoA, iron, and dopamine metabolic defects in mammalian models of PKAN

Suh Young Jeong[1] (iD), Penelope Hogarth[1,2], Andrew Placzek[3], Allison M Gregory[1], Rachel Fox[1] (iD), Dolly Zhen[1], Jeffrey Hamada[1], Marianne van der Zwaag[4], Roald Lambrechts[4], Haihong Jin[3], Aaron Nilsen[3], Jared Cobb[5], Thao Pham[5], Nora Gray[2], Martina Ralle[1], Megan Duffy[1], Leila Schwanemann[1], Puneet Rai[1], Alison Freed[1], Katrina Wakeman[1], Randall L Woltjer[5], Ody CM Sibon[4] (iD) & Susan J Hayflick[1,2,6,*] (iD)

## Abstract

**Pantothenate kinase-associated neurodegeneration (PKAN) is an inborn error of CoA metabolism causing dystonia, parkinsonism, and brain iron accumulation. Lack of a good mammalian model has impeded studies of pathogenesis and development of rational therapeutics. We took a new approach to investigating an existing mouse mutant of *Pank2* and found that isolating the disease-vulnerable brain revealed regional perturbations in CoA metabolism, iron homeostasis, and dopamine metabolism and functional defects in complex I and pyruvate dehydrogenase. Feeding mice a CoA pathway intermediate, 4′-phosphopantetheine, normalized levels of the CoA-, iron-, and dopamine-related biomarkers as well as activities of mitochondrial enzymes. Human cell changes also were recovered by 4′-phosphopantetheine. We can mechanistically link a defect in CoA metabolism to these secondary effects via the activation of mitochondrial acyl carrier protein, which is essential to oxidative phosphorylation, iron–sulfur cluster biogenesis, and mitochondrial fatty acid synthesis. We demonstrate the fidelity of our model in recapitulating features of the human disease. Moreover, we identify pharmacodynamic biomarkers, provide insights into disease pathogenesis, and offer evidence for 4′-phosphopantetheine as a candidate therapeutic for PKAN.**

**Keywords** 4′-phosphopantetheine; coenzyme A; NBIA; PANK2; PKAN
**Subject Categories** Neuroscience; Pharmacology & Drug Discovery
See also: **RA Lambrechts et al** (December 2019)

## Introduction

Pantothenate kinase-associated neurodegeneration (PKAN) is an autosomal recessive movement disorder affecting children and adults. This profoundly disabling disorder manifests with severe, painful dystonia, young-onset parkinsonism, globus pallidus iron accumulation, and blindness from pigmentary retinopathy (Hayflick *et al*, 2003). PKAN is one of the neurodegeneration with brain iron accumulation (NBIA) disorders and though ultra-rare, is readily identifiable in the clinical setting by its distinctive brain MRI pattern, the "eye of tiger" sign. As disease advances, affected persons lose control of movement yet retain substantial intellectual function. Currently, no disease-modifying therapeutic is available.

Pantothenate kinase-associated neurodegeneration is an inborn error of coenzyme A (CoA) metabolism that may be amenable to a therapeutic approach in which the enzymatic defect is bypassed by supplying a "downstream" pathway intermediate (Fig 1A; Zhou *et al*, 2001). Pantothenate kinase catalyzes the first step in *de novo* CoA synthesis starting from vitamin B₅ (pantothenate, Fig 1A), a function in mammals that is shared by four isozymes (Leonardi *et al*, 2005). PKAN is caused by mutations in *PANK2*, encoding pantothenate kinase 2 (Zhou *et al*, 2001), which is the only isozyme that localizes to mitochondria (Hortnagel *et al*, 2003; Johnson *et al*, 2004; Kotzbauer *et al*, 2005). CoA is essential for hundreds of metabolic reactions including the tricarboxylic acid cycle, fatty acid oxidation and synthesis, amino acid metabolism, and neurotransmitter synthesis (Strauss, 2010). Moreover, the ratio of CoA to acetyl-CoA is a central determinant in coordinating cellular metabolism with gene regulation (Pietrocola *et al*, 2015). Despite its importance at the nexus of intermediary metabolism, our knowledge of CoA

1 Department of Molecular & Medical Genetics, Oregon Health & Science University, Portland, OR, USA
2 Department of Neurology, Oregon Health & Science University, Portland, OR, USA
3 Medicinal Chemistry Core, Oregon Health & Science University, Portland, OR, USA
4 Department of Cell Biology, University Medical Center Groningen, Groningen, the Netherlands
5 Department of Pathology, Oregon Health & Science University, Portland, OR, USA
6 Department of Pediatrics, Oregon Health & Science University, Portland, OR, USA
*Corresponding author. Tel: +1 (503) 494 7703; E-mail: hayflick@ohsu.edu

synthesis, homeostasis, and transport is incomplete, as is our knowledge of the mechanism by which loss of pantothenate kinase 2 leads to neurodegeneration and iron accumulation. The canonical CoA synthesis pathway (Fig 1A) was thought to be the only source of cellular CoA until recent work revealed an alternate mechanism to synthesize CoA within cells from the extracellular delivery of the pathway intermediate 4′-phosphopantetheine (Srinivasan *et al*, 2015). Srinivasan *et al* further reasoned that 4′-phosphopantetheine may have therapeutic potential in PKAN to bypass the pantothenate kinase 2 defect and restore cellular CoA synthesis.

To test this idea, we needed to develop a high-fidelity mammalian model of PKAN. Published mouse models rely on compound abnormalities in order to demonstrate a phenotype, requiring both the deletion of *Pank2* and a superimposed second "hit", either genetic or metabolic. They include (i) a neuron-specific *Pank1+Pank2* double knock-out model (Sharma *et al*, 2018); (ii) a *Pank2* knock-out animal fed a severe ketogenic diet to induce metabolic stress (Brunetti *et al*, 2014); and (iii) mice administered hopantenate, a toxic chemical that competes with pantothenate as a substrate for all pantothenate kinases and causes global depletion of CoA with lethal metabolic changes (Zhang *et al*, 2007; Di Meo *et al*, 2017). A fundamental limitation of all three models is the inability to attribute disease features specifically to defective pantothenate kinase 2. As a result, hypotheses of PKAN pathogenesis based on these models are uncertain, and therapeutics developed using these models may or may not have efficacy in PKAN (Brunetti *et al*, 2014; Di Meo *et al*, 2017; Sharma *et al*, 2018).

We sought to develop a mammalian disease model with features that could be specifically attributed to loss of pantothenate kinase 2. Employing knowledge of the human disease, we re-investigated our previously reported mice, which harbor a germline null mutation in *Pank2* and have no detectable pantothenate kinase 2 protein (Kuo *et al*, 2005). Though the animals manifested a mild, late-onset retinopathy and pupillometric defect, similar to features found in humans (Hayflick *et al*, 2003; Egan *et al*, 2005), these features were strain-specific and required electroretinographic expertise to track (Kuo *et al*, 2005). Those limitations, coupled with the long duration to a clinical phenotype and lack of overt neurological features, appeared to restrict the utility of this mutant. We returned to these animals to determine whether disease changes might be detectable if we isolated the disease-vulnerable globus pallidus region from the remaining brain tissue and then looked for CoA-related differences. This idea was based on knowledge of the exquisitely focal human neuropathology, which is limited to globus pallidus (Woltjer *et al*, 2015). Our new approach yielded success, revealing critical insights into disease pathogenesis by demonstrating defects in CoA metabolism, iron homeostasis, dopamine metabolism, and mitochondrial function in globus pallidus. Moreover, we used this model to demonstrate that 4′-phosphopantetheine ameliorates the primary CoA metabolic defect and normalizes all secondary perturbations.

## Results

### Enriching for disease-vulnerable brain tissue reveals CoA pathway defects in *Pank2* KO animals

Since PKAN selectively damages globus pallidus, we sought to isolate this disease-vulnerable region from other brain tissue in the

*Pank2* KO mouse for further investigation. We dissected mouse brain into three regions: globus pallidus-containing (GP), substantia nigra-containing (SN), and cerebellum (Fig 1B). GP also includes thalamus, hypothalamus, and striatum. SN also includes ventral tegmental area, red nucleus, and oculomotor nucleus. The method of dissection was confirmed for each region based on gene expression patterns. We found candidate genes using *in situ* hybridization data reported in the Allen Brain Atlas (©2016 Allen Institute for Brain Science. Allen Mouse Brain Atlas. Available from: mouse.brain-map.org) and confirmed high levels of mRNA for *Drd1* in GP (but not SN or cerebellum), *Th* in SN (but not GP or cerebellum), and *Gabra6* and *Calb1* in cerebellum (but not GP or SN) using qRT–PCR (Appendix Fig S1B).

With this new approach, we set out to determine whether we could identify perturbations in the CoA pathway and in disease-relevant biomarkers. Using the three brain regions from WT and KO animals, we measured mRNA expression for the three genes encoding CoA synthetic enzymes that are downstream of pantothenate kinase (Fig 1A), including *Ppcs* (phosphopantothenoylcysteine synthetase), *Ppcdc* (phosphopantothenoylcysteine decarboxylase), and *Coasy* (CoA synthase). The expression of two, *Ppcs* and *Coasy,* was significantly down-regulated in KO animals but only in the GP region (Fig 1C). *Ppcdc* mRNA expression did not differ by genotype (Fig 1C) and was not studied further. Levels of Coasy protein were also found to be decreased in KO GP only (Fig 1D). For this reason and because it is the terminal enzyme required for CoA synthesis, we considered *Coasy* expression as a candidate biomarker for further development.

### Defective Pank2 perturbs iron homeostasis, mitochondrial function, and dopamine metabolism

A common feature among the NBIA disorders is iron accumulation in globus pallidus. To assess iron homeostasis in our model, we measured the expression of iron homeostasis genes, levels of subcellular compartment iron, and activity of an iron-dependent enzyme. The expression of *Tfrc* (transferrin receptor 1), *Ireb2* (iron regulatory protein 2), and *Hamp* (hepcidin) was significantly decreased in KO animals in GP only (Fig 2A), and Tfr1 protein levels were also markedly decreased (Fig 2F). These findings suggested that cells in this region were sensing and responding to increased cytosolic iron. We confirmed the presence of significantly increased iron levels in cells isolated from GP in the KO animals in both the cytosolic and mitochondrial fractions using subcellular fractionation and inductively coupled plasma mass spectroscopy (Fig 2B). In contrast, iron levels in cortex and SN subcellular fractions did not differ by genotype (Appendix Fig S2A). We confirmed that there were equivalent quantities of mitochondria in tissue samples from KO and WT GP using mitochondrial DNA quantification (data not shown).

We then sought to determine whether the changes in CoA metabolism and iron homeostasis alter the function of enzymes that depend on these factors. Mitochondrial aconitase requires iron–sulfur cluster biogenesis for its activity and also depends on CoA for production of its substrate, citrate. Mitochondrial aconitase catalyzes the isomerization of citrate to isocitrate in the tricarboxylic acid cycle, and citrate is formed by the condensation of acetate, derived from acetyl-CoA, and oxaloacetate. We found loss of activity of mitochondrial aconitase in KO brain but not in WT brain or liver

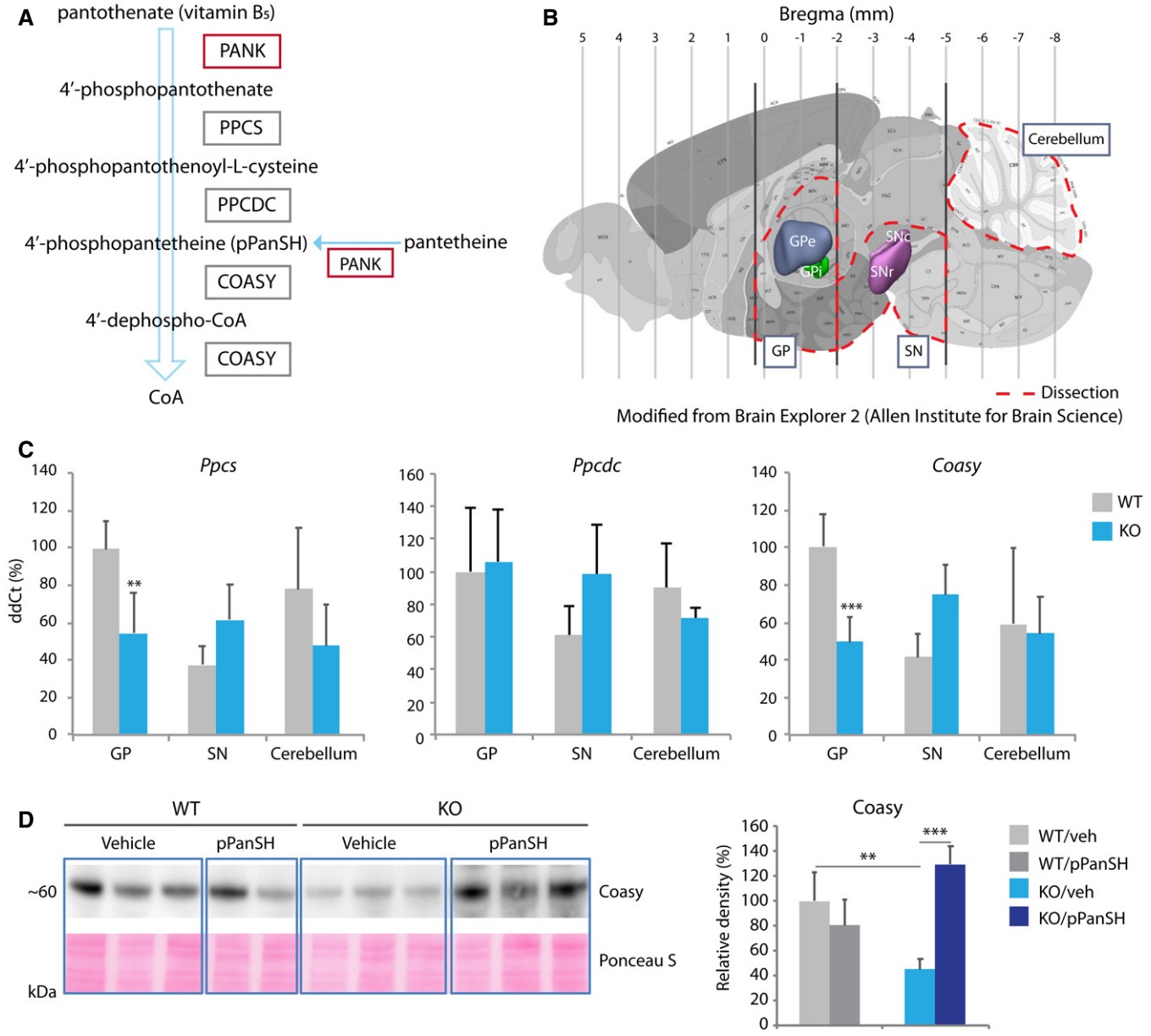

**Figure 1. Isolating disease-vulnerable brain tissue from disease-protected reveals CoA pathway defects in *Pank2* KO animals.**

A  The CoA synthesis pathway, including intermediates and enzymes (boxed).

B  Brain dissection yielding three study regions (red dotted lines) including GP, SN, and cerebellum. Captured and modified from Brain Atlas 2 (Allen Institute for Brain Science).

C  Relative quantification of mRNA for *Ppcs, Ppcdc,* and *Coasy* by genotype from each of the three study regions. $n$ = 11, 4, 4 for WT and $n$ = 8, 4, 4 for KO (GP, SN, Cerebellum, respectively).

D  Western blot and quantification for Coasy from GP by genotype and treatment status (pPanSH = 4′-phosphopantetheine). Ponceau S was used as a loading control. $n$ = 8, 4 for WT and $n$ = 8, 6 for KO (vehicle and pPanSH groups, respectively).

Data information: Data shown here were evaluated by either one-way ANOVA or two-way ANOVA depending on the number of variables. **$P < 0.01$, ***$P < 0.001$. All graphs represent mean ± s.e.m.

from either genotype (Fig 2C, Appendix Fig S2B). Previous studies in human PKAN iPSC-derived neurons also reported perturbations in aconitase and iron homeostasis, but they differ from what we observed. Specifically, Orellana *et al* (2016) reported decreased activities of both mitochondrial aconitase and cytosolic aconitase as well as TfR1 up-regulation and FtH (ferritin) down-regulation,

suggesting that the iPSC-derived neurons were sensing iron insufficiency. Reasons for these differences in the different systems are uncertain.

We sought further evidence for functional defects that could be attributed to CoA and iron dyshomeostasis. The synthesis of acetyl-CoA requires pyruvate dehydrogenase (PDH) and depends on

sufficient quantities of mitochondrial matrix CoA. We found significantly decreased PDH activity from GP in KO animals compared with controls, with no accompanying loss of protein (Fig 2D and F, Appendix Fig S2C). Because iron is essential for electron transport chain function, we also measured complex I activity and found a significant decrease in activity in GP from the KO mouse compared to WT controls (Fig 2E). Our combined data show that defective

*Pank2* causes regional alterations in the expression and activity of a specific set of enzymes related to CoA metabolism and iron homeostasis with functional sequelae in energy metabolism.

Pantothenate kinase-associated neurodegeneration manifests with dystonia and parkinsonism; however, a specific defect in dopamine metabolism or transport has not been shown. In the GP-enriched region of KO animals, we saw marked up-regulation in

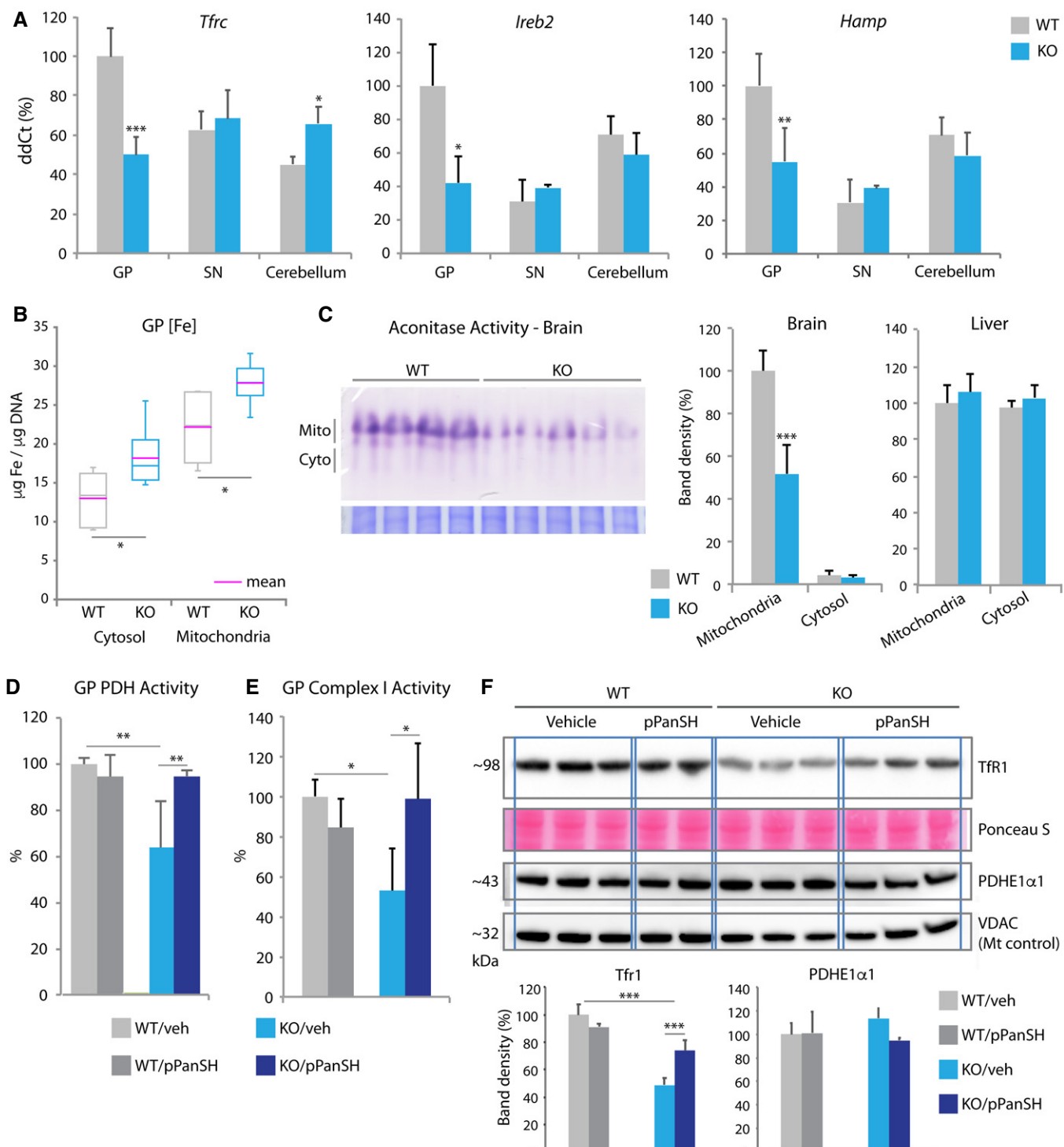

**Figure 2.**

**Figure 2. Regional brain differences in iron homeostasis suggest a mechanism for iron overload in PKAN.**

A   Relative mRNA expression of *Tfrc, Ireb2,* and *Hamp* by genotype and brain region. *n* = 11, 4, 4 for WT and *n* = 8, 4, 4 for KO (GP, SN, cerebellum, respectively).

B   Total iron quantity in cytosol and mitochondria from GP using ICP-MS. The data distribution is presented using a box plot, including minimum, first quartile, median, third quartile, and maximum. The magenta-colored line represents mean. *n* = 6 for both genotypes.

C   Assay of mitochondrial versus cytosolic aconitase activity in brain compared by genotype. Quantification of aconitase activity in two tissues (brain and liver) from animals of each genotype by band density. *n* = 4 for WT, *n* = 5 for KO.

D, E   Activity of pyruvate dehydrogenase (D) and complex I (E) in GP by genotype and treatment status (veh = vehicle; pPanSH = 4′-phosphopantetheine). *n* = 5 for both genotypes and treatment groups.

F   Western blot and quantification of TfR1 and PDHE1α1 from GP by genotype and treatment status (pPanSH = 4′-phosphopantetheine). Ponceau S was used as total protein loading control and VDAC for the mitochondrial protein loading control. *n* = 6, 4 for WT and *n* = 6, 6 for KO (vehicle and pPanSH groups, respectively).

Data information: Data were evaluated by one-way ANOVA or two-way ANOVA depending on the number of variables. *$P < 0.05$, **$P < 0.01$, ***$P < 0.001$. All graphs except (B) represent mean ± s.e.m.

dopamine receptor gene expression (*Drd1* and *Drd2*, Fig 3A and B) and in Drd1 protein levels (Fig 3A, Drd2 protein was not studied). In contrast, GABA receptor gene expression (*Gabra3* and *Gabra6*) was not altered in any of the three brain regions tested in KO animals (Fig 3C, Appendix Fig S3A). Receptor up-regulation commonly results from loss of ligand. Therefore, we looked for evidence of diminished dopamine synthesis by measuring levels of tyrosine hydroxylase, the rate-limiting enzyme in the dopamine synthesis pathway, and one requiring iron as a cofactor. Decreased Th protein levels were found in whole brain from *Pank2* KO animals (Fig 3D), but the mice show no signs of parkinsonism. More refined studies to detect subtle evidence for dystonia are planned. In

summary, we propose that loss of pantothenate kinase 2 disrupts dopamine homeostasis, causing the movement disorder observed in humans.

Within mouse brain, we see a regional pattern of disease changes that mirror those found in human brain. We therefore expected that non-diseased tissue from other organ systems would not manifest the biochemical changes we found in globus pallidus. To our surprise, both cultured human fibroblasts and immortalized and fresh lymphocytes from people with PKAN show defects in the CoA pathway. Using primary cells, we corroborated our mouse findings; *COASY* expression was decreased in cultured human fibroblasts and lymphoblasts (Fig 4A), as was *PPCS* expression (Appendix Fig S4A).

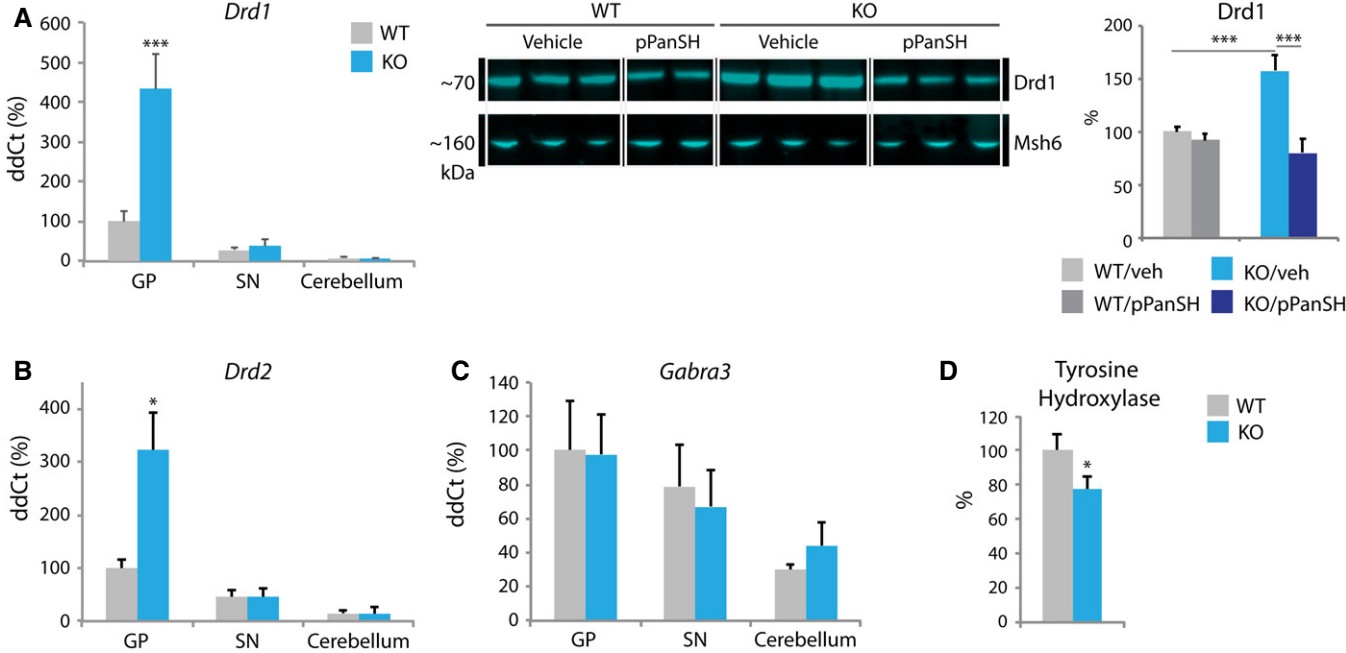

**Figure 3. Dopamine metabolism is impaired in *Pank2* KO GP.**

A   Relative mRNA and protein expression of the dopamine receptor *Drd1* compared by genotype, brain region, and treatment status (pPanSH = 4′-phosphopantetheine). *n* = 11, 4, 4 for WT and *n* = 8, 4, 4 for KO (GP, SN, cerebellum, respectively) qRT–PCR. *n* = 5 for Western blot for all groups.

B, C   Relative mRNA expression of *Drd2* and GABA receptor *Gabra3* expression by genotype and brain region. *n* = 11, 4, 4 for WT and *n* = 8, 4, 4 for KO (GP, SN, cerebellum, respectively).

D   Western blot quantification from whole brain of tyrosine hydroxylase compared by genotype. *n* = 5 for WT and *n* = 3 for KO.

Data information: Statistical significance was determined using ANOVA. *$P < 0.05$, ***$P < 0.001$. All graphs represent mean ± s.e.m.

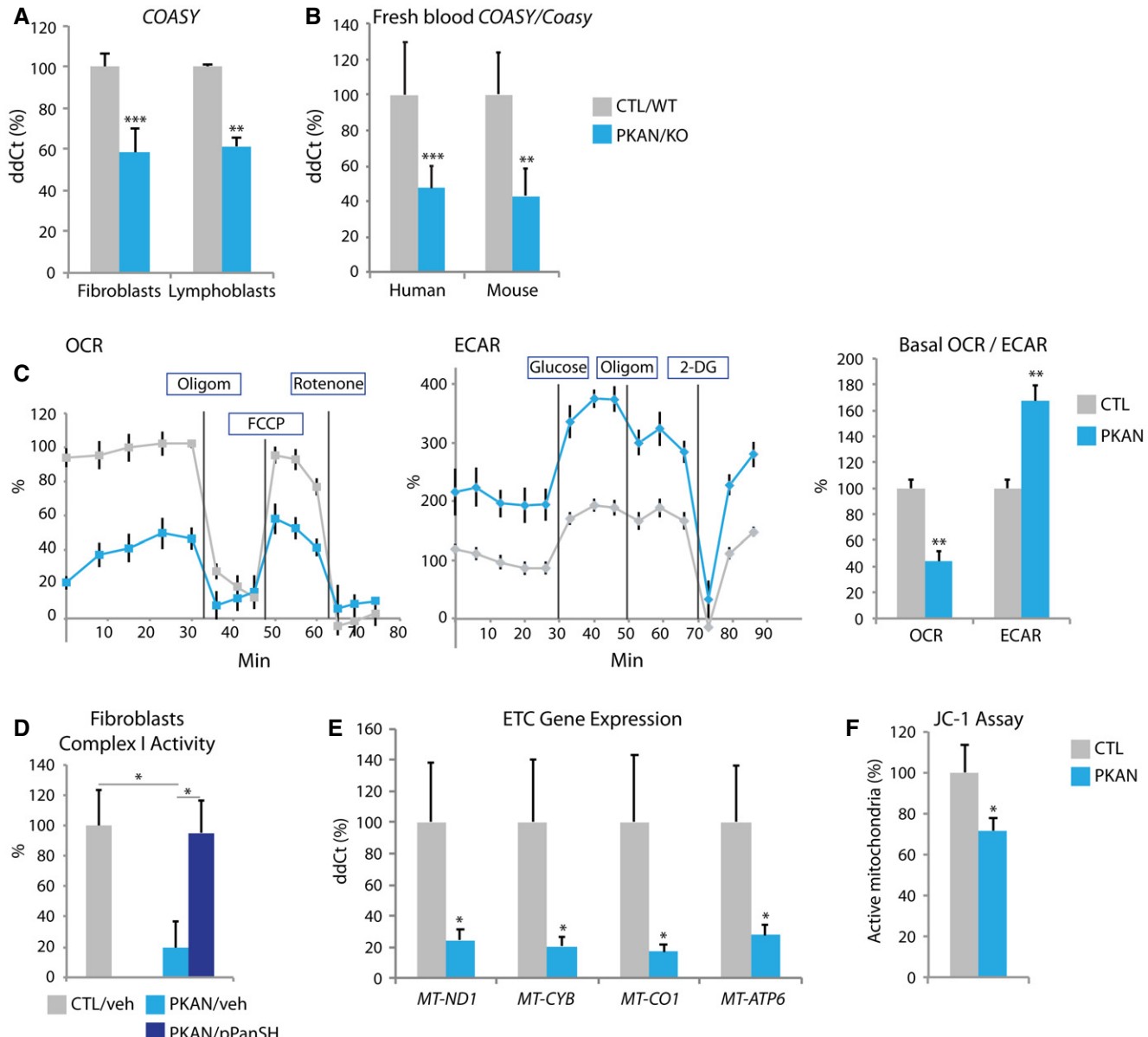

**Figure 4. Human cells with *PANK2* mutations show perturbations in the CoA synthesis pathway and diminished mitochondrial respiration.**

A   Relative quantification of *COASY* mRNA in human PKAN primary cell lines. *n* = 5 for both genotypes and cell types.

B   Relative quantification of *COASY/Coasy* in fresh, blood-derived lymphocytes from human and mouse, respectively compared by genotype. *n* = 51, 35 for human (control, PKAN, respectively) and *n* = 5 for both mouse genotypes.

C   Extracellular flux analysis in human fibroblasts showing differences in OCR (oxygen consumption rate) and ECAR (extracellular acidification rate) by genotype. Oligom; oligomycin, FCCP; carbonyl cyanide-p-trifluoromethoxyphenylhydrazone, 2-DG; 2-deoxy glucose. Quantification of baseline OCR and ECAR is shown, as well. *n* = 3 per genotype with four technical replicates.

D   Complex I activity assay using fibroblast lysates. Cells were either treated with vehicle or 50 μM pPanSH for 3 days, and total protein extract was used for the Complex I Dipstick assay *n* = 4 for both genotypes and treatment groups.

E   Relative mRNA expression of mitochondrially encoded genes important for oxidative phosphorylation from complex I (*MT-ND1*), complex III (*MT-CYB*), cytochrome C (*MT-CO1*), and ATP synthase (*MT-ATP6*) in human mutant fibroblasts versus control cells. *n* = 3 for both genotypes.

F   Calculation of active mitochondria using the JC-1 assay by genotype in control and PKAN lymphoblasts. *n* = 2.

Data information: Data were analyzed using one-way or two-way ANOVA for statistical significance. *$P < 0.05$; **$P < 0.01$, ***$P < 0.001$. All graphs represent mean ± s.e.m.

Differences by genotype were also found in circulating cells. Untransformed fresh blood-derived lymphocytes from people with PKAN showed decreased expression of *COASY* compared to age-matched controls, and similar changes were seen in mice (Fig 4B and Appendix Fig S6B). Furthermore, we demonstrated defects in mitochondrial function in patient-derived fibroblasts, as well. Using

extracellular flux analysis of cultured primary fibroblasts from PKAN patients, we measured oxygen consumption rate (OCR) as a proxy for respiration and extracellular acidification rate (ECAR) as a surrogate for glycolysis. We found decreased OCR and increased ECAR in PKAN cells compared with control cells (Fig 4C), adding to previously published work done on iPSC-derived neurons (Santambrogio *et al*, 2015). Complex I activity in PKAN fibroblasts was decreased significantly, as well (Fig 4D). Mitochondrial aconitase is known to serve a distinct function in coordinating nuclear and mitochondrial gene expression (Chen *et al*, 2005). Thus, we asked if expression of mitochondrially encoded genes might be affected in these cells. Indeed, we found decreased expression of genes encoding subunits for complexes I, II, IV, and V (Fig 4E), suggesting a further basis for reduced respiration. Impaired oxidative phosphorylation would lead to loss of the mitochondrial membrane potential, which we confirmed in patient-derived cells using the JC-1 assay (Fig 4F). The rise in ECAR suggests that patient cells had undergone a glycolytic shift, which may also explain the metabolic perturbations that we observed in mouse brain. Loss of respiratory capacity can lead to a shift toward glycolysis for energy production. Thus, the human PKAN cell data replicated and expanded results from the KO mouse, corroborating defects in CoA metabolism and dysfunction of mitochondria.

We have identified a set of biomarkers that reflect defects in CoA synthesis, iron homeostasis, mitochondrial function, and dopamine metabolism in experimental mammalian models of PKAN. These perturbations mirror key features of the human disease and suggest that our models can be used to understand PKAN pathogenesis and to investigate candidate therapeutics.

### PPARα agonists fail to rescue brain CoA pathway defects

Why is the globus pallidus selectively vulnerable to loss of pantothenate kinase 2 function? We considered whether differences in compensation by other pantothenate kinase proteins might explain regional CoA pathway defects in brain. We analyzed *Pank1* and *Pank3* expression as a function of brain region and *Pank2* genotype. *Pank1* produces two protein isoforms, Pank1α and Pank1β, which differ in their transcriptional and enzymatic regulation (Rock *et al*, 2000, 2002) (Ramaswamy *et al*, 2004). We identified a paradoxical decrease in *Pank1α* expression only in KO GP compared to WT, with no change in *Pank1β* expression (Fig 5A). We found no differences in other brain regions or *Pank3* expression (Fig 5A and B).

This observation suggested a possible therapeutic approach to compensate for Pank2 loss via *Pank1α* up-regulation. Previously, selective up-regulation of human *PANK1α* transcription by a PPARα agonist, bezafibrate, was reported in cultured hepatoblastoma cells (Ramaswamy *et al*, 2004). We asked whether the PPARα agonists bezafibrate and gemfibrozil might normalize *Pank1α* expression in GP in the KO animals. Oral dosing for 14 days with bezafibrate (0.8 mg/g body weight) and gemfibrozil (1.2 mg/g body weight) failed to increase expression of either *Pank1α* or *Coasy* in GP in KO animals (Fig 5C). To confirm that each compound was consumed and reached brain, we analyzed expression of a gene known to be up-regulated by PPARα agonists (Chen *et al*, 2017). We quantified expression in GP of *Cpt1c*, encoding a neuronal isoform of carnitine palmitoyltransferase (Chen *et al*, 2017), and demonstrated that both

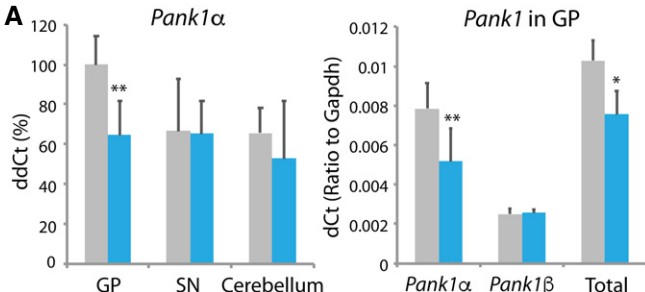

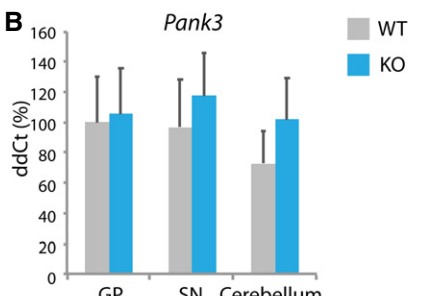

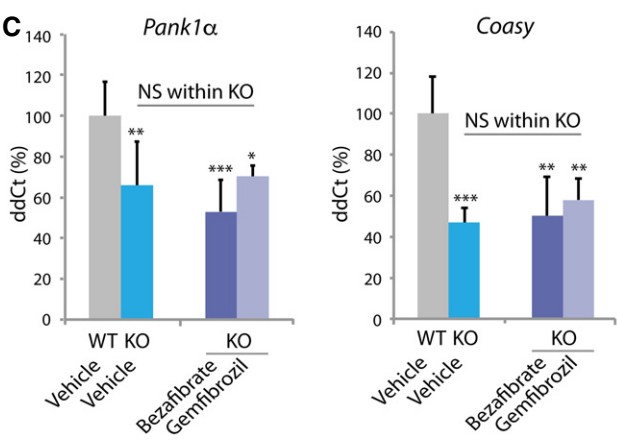

**Figure 5. *Pank1* expression differences and recovery attempt using PPARα agonists.**

A, B   Relative expression of *Pank1α*, *Pank1β*, total *Pank1*, and *Pank3* in mouse brain regions. *n* = 8, 8, 4 for both genotypes (GP, SN, cerebellum, respectively).

C   Relative expression of *Pank1α* and *Coasy* in GP from WT and KO mice treated for 14 days with bezafibrate (0.8 mg/g body weight) or gemfibrozil (1.2 mg/g body weight). *n* = 8 for both genotypes treated with vehicle, *n* = 5 for both KO groups treated with either bezafibrate or gemfibrozil.

Data information: Data were evaluated by one-way or two-way ANOVA depending on the number of variables. *$P < 0.05$, **$P < 0.01$, ***$P < 0.001$. All graphs represent mean ± s.e.m.

fibrate compounds up-regulated expression of *Cpt1c* in GP from animals of both genotypes, confirming that the compound altered the expression of other genes in brain as expected (Appendix Fig S5B). These results suggested that *Pank1* expression may be regulated differently by tissue type or species or both. Alternatively, a primary defect in Pank2 or its metabolic sequelae might block PPARα up-regulation of *Pank1α* expression. Regardless, this therapeutic approach proved unlikely to yield benefit in PKAN.

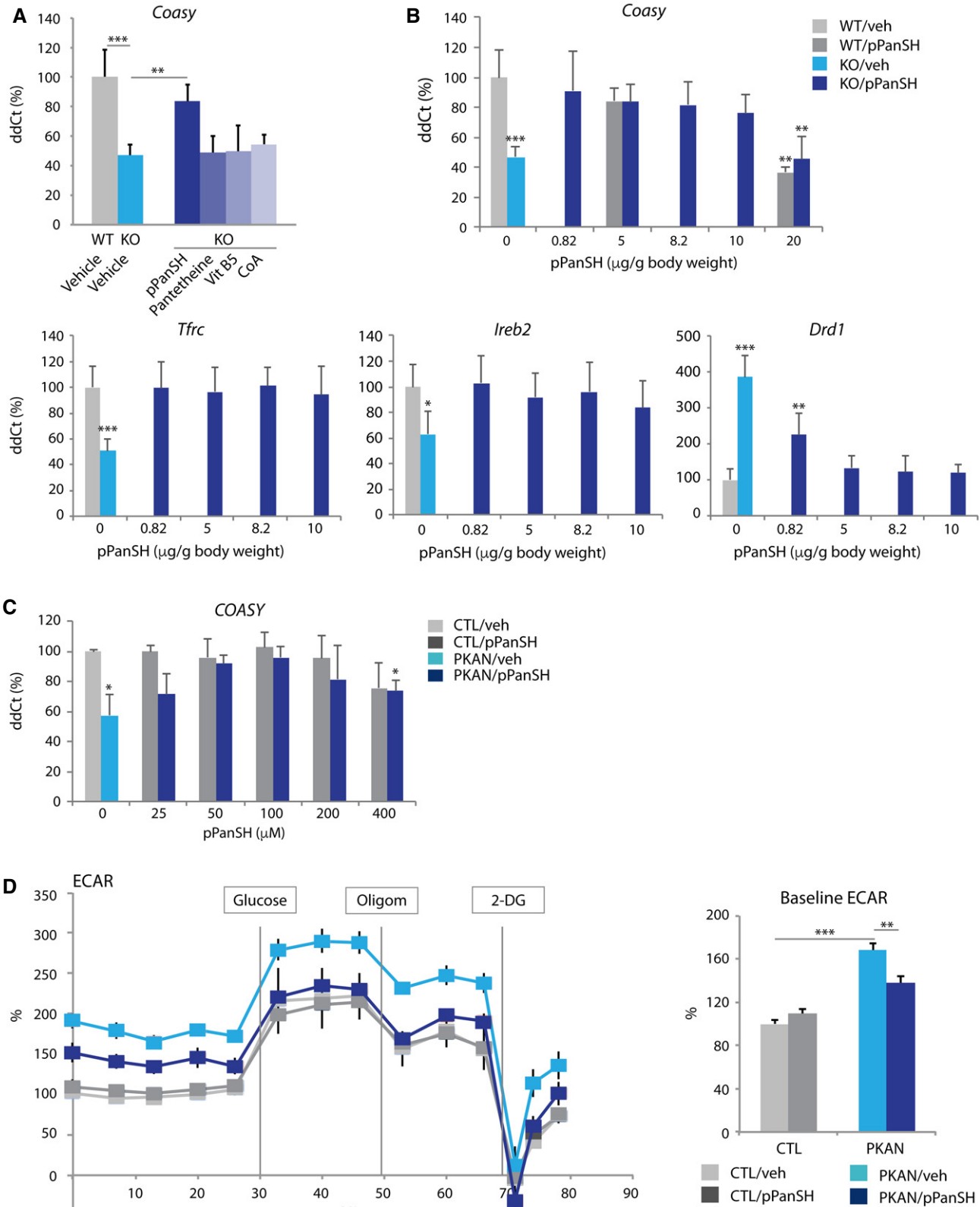

Figure 6.

Figure 6.   4′-Phosphopantetheine corrects markers of CoA, iron, and dopamine dyshomeostasis and mitochondrial dysfunction in *Pank2* KO mice.

A   Relative quantification of *Coasy* mRNA from GP compared by treatment status using four compounds: 4′-phosphopantetheine (pPanSH), pantetheine, vitamin B₅, and CoA. *n* = 8 (WT-veh), 5 (KO-veh), 5 (KO-pPanSH), 4 (KO- other three compounds).

B   Correction of *Coasy, Tfrc, Ireb2,* and *Drd1* expression in GP by 4′-phosphopantetheine over a range of doses. *n* = 8 (WT-veh), 5 (KO-veh), 5 (KO-0.82), 4 (WT-5), 5 (KO-5), 5 (KO-8.2), 5 (KO-10). 5 (WT-20), 5 (KO-20).

C   Relative quantification of *COASY* mRNA from human fibroblasts treated with pPanSH. *n* = 3 for both genotypes.

D   ECAR measured in cultured human fibroblasts by genotype and treatment status. Cells were treated for 3 days with 4′-phosphopantetheine (pPanSH) or vehicle (veh). *n* = 3 for both genotypes.

Data information: Data were evaluated by one-way or two-way ANOVA depending on the number of variables. Asterisks shown in (B–D) represent statistical significance compared to the vehicle-treated control groups. *$P < 0.05$, **$P < 0.01$, ***$P < 0.001$. All graphs represent mean ± s.e.m.

### 4′-phosphopantetheine normalizes CoA, iron, and dopamine biomarkers as well as complex I and PDH activities

Using our mouse model, we asked whether the CoA metabolic defect could be corrected by providing alternate substrates, thereby bypassing the pantothenate kinase defect. We administered equimolar doses of pantetheine, 4′-phosphopantetheine, or CoA orally to mice for 14 days and then analyzed PKAN-related biomarkers. Pantothenate (vitamin B₅) was administered as a control. Only 4′-phosphopantetheine normalized *Coasy* expression, suggesting that *Coasy* expression is inducible by substrate availability (Fig 6A). *Ppcs*, which encodes an enzyme upstream of 4′-phosphopantetheine, was predictably unchanged by the administration of 4′-phosphopantetheine or any of the other compounds tested (data not shown). We observed a dose–responsive normalization in expression of *Coasy, Tfrc, Ireb2,* and *Drd1* (Fig 6B) and correction of protein levels of Coasy, Tfr1, and Drd1 in KO GP (Figs 1D and 2F, and 3A). Even the lowest dose of 4′-phosphopantetheine (0.82 μg/g) was effective in fully correcting the CoA- and iron-related gene abnormalities. In addition, PDH and complex I activities in KO GP were fully recovered (Fig 2D and E). These data suggest that 4′-phosphopantetheine is orally bioavailable and crosses the blood–brain barrier. With higher doses of 4′-phosphopantetheine, we observed progressively lower levels of *Coasy* expression in GP from both WT and KO animals (Fig 6B), suggesting that *Coasy* expression is repressible by product inhibition. *Coasy* expression in KO mouse circulating lymphocytes was also normalized by 14 days of administration of 4′-phosphopantetheine dosed at 5 μg/g, and we observed the same inverse dose response seen at higher doses in both WT and KO animals (Appendix Fig S6B). While an alternate hypothesis to be considered is that lower levels of 4′-phosphopantetheine alleviate basal repression in the KO animal, this idea does not explain similar observations in WT animals. Therefore, we favor the more parsimonious interpretation of product inhibition to explain all results. Our data indicate that 4′-phosphopantetheine corrects the primary CoA pathway-related biomarkers and normalizes secondary defects in iron, dopamine, complex I, and PDH, a striking finding.

Human cells showed rescue with 4′-phosphopantetheine treatment, as well. Primary PKAN fibroblasts treated for 24 h normalized their expression of *COASY* and *TFRC* in a dose-dependent manner (Fig 6C and Appendix Fig S6C). We again observed an inverse dose response in *COASY* expression at higher doses suggesting downregulation as a result of product inhibition. 4′-phosphopantetheine decreased the glycolytic shift and normalized complex I activity, indicating improved mitochondrial respiration in this *in vitro* human model (Figs 4D and 6D) and corroborating our findings of functional correction in mouse brain.

We evaluated the duration of effect of 4′-phosphopantetheine on brain biomarkers after its withdrawal. As before, we administered 4′-phosphopantetheine orally for 14 days at 5 μg/g and then ceased administration and sacrificed animals at 0, 1, 2, 3, and 7 days postcessation. We found that expression in GP of *Coasy, Tfrc,* and *Drd1* all drifted back to pre-treatment levels in a time-dependent manner over 7 days (Fig 7A). These results suggest that 4′-phosphopantetheine can be titrated to correct the biomarker abnormalities resulting from loss of pantothenate kinase 2, albeit with the need to sacrifice animals for the analysis. In this way, the earliest pathophysiologic changes in PKAN can be investigated in a highly tractable true model system.

We found no evidence for toxicity of 4′-phosphopantetheine. A dose of 20 μg/g was administered orally for 14 days to wild-type mice. Histologic analysis of brain, spinal cord, heart, muscle, liver, kidney, and spleen by two pathologists blinded to treatment status revealed no differences (data not shown).

In summary, our data showed perturbations in a set of disease-relevant biomarkers in mouse and human models of PKAN that were corrected by 4′-phosphopantetheine. Defective pantothenate kinase 2 resulted in a CoA pathway defect and secondary perturbations in iron homeostasis, dopamine metabolism, complex I and PDH activities, and mitochondrial respiration. From these, we have identified candidate pharmacodynamic biomarkers. Our data showed that 4′-phosphopantetheine corrected the CoA-related defects and resolved all secondary abnormalities in both models. With these results, we have demonstrated the fidelity of our mouse model in recapitulating key features of the human disease and provided evidence in support of 4′-phosphopantetheine as a candidate therapeutic for PKAN.

## Discussion

We present a mouse model of PKAN with CoA, iron, and dopamine metabolic defects that can be specifically attributed to loss of pantothenate kinase 2 function. This model recapitulates key features of the human disease, including iron accumulation and brain region specificity, and therefore represents a tractable system with which to investigate disease pathogenesis and evaluate candidate therapeutics.

High value has been placed on mouse models with features that mimic the clinical manifestations of disease in humans. Animals showing only biochemical and molecular changes have garnered less favor as models, even when those changes are specifically attributable to the primary cause of disease, as in single-gene disorders. In fact, such models are likely to be more informative for

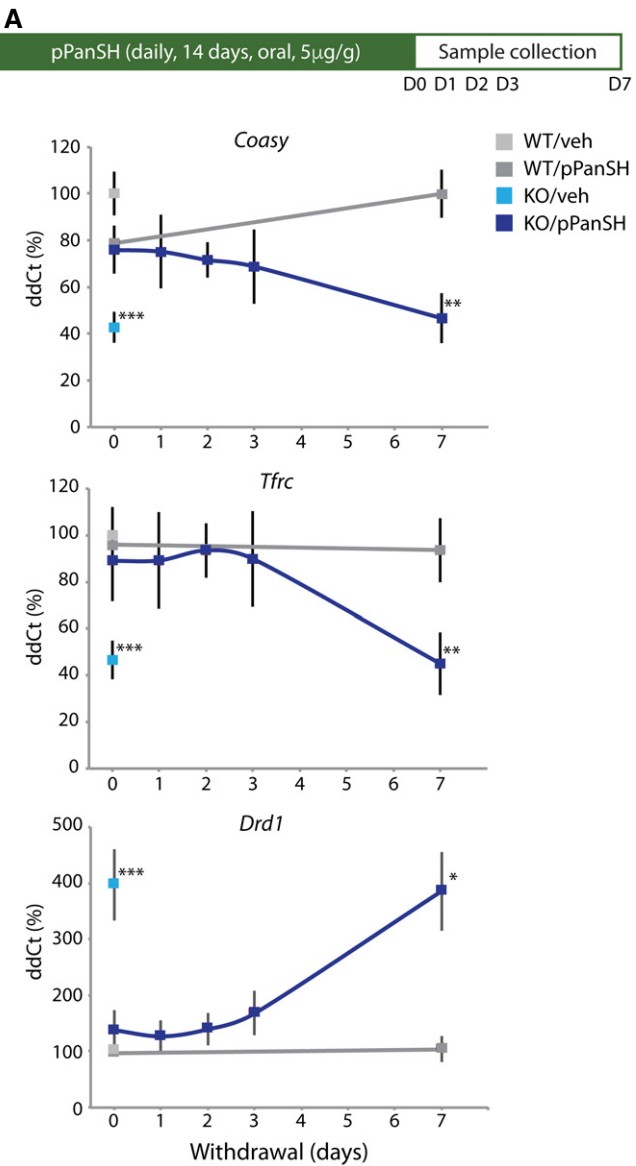

**Figure 7. Withdrawal of 4′-phosphopantetheine causes brain biomarkers to return to pre-treatment levels.**

Relative quantification of *Coasy*, *Tfrc*, and *Drd1* mRNA from GP by number of days post-cessation of treatment with 5 μg of 4′-phosphopantetheine per g of body weight. *n* = 11 (WT-veh-D0), 5 (KO-veh-D0), 4 (WT-pPanSH-D0), 5 (KO-pPanSH-D0), 5 (KO-pPanSH-D1), 5 (KO-pPanSH-D2), 5 (KO-pPanSH-D3), 5 (WT-pPanSH-D7), 4 (KO-pPanSH-D7). Data shown were evaluated by one-way ANOVA. Asterisks represent statistical analyses compared to the WT vehicle group. *$P$ < 0.05, **$P$ < 0.01, ***$P$ < 0.001. All graphs represent mean ± s.e.m.

investigating critical early disease changes before cells and tissues sustain extensive damage. Molecular perturbations precede clinical manifestations and are arguably more informative than an overt neurological sign. We now report *Pank2* KO animals with abnormal levels of CoA-, iron-, and dopamine-related biomarkers, diminished activities of complex I and pyruvate dehydrogenase, and mitochondrial dysfunction, all of which were found to be regionally localized in brain. These changes represent the molecular cascade of early PKAN pathogenesis resulting from loss of pantothenate kinase 2.

We favor a hypothesis of PKAN pathogenesis proposed by Lambrechts *et al* (2019) that integrates and explains the myriad cellular changes we report (Fig 8). If there is a deficiency of CoA in PKAN, processes in which there is net consumption of CoA may be most sensitive to that loss. CoA is consumed for the phosphopantetheinyl activation of certain proteins, whereas most other pathways that utilize CoA result in no net change in levels. These activated proteins play essential roles in mammals, including as acyl carrier proteins (Praphanphoj *et al*, 2001; Donato *et al*, 2007). Mitochondrial acyl carrier protein (mtACP), which requires phosphopantetheinyl activation, is essential for electron transport, type II fatty acid synthesis, iron–sulfur cluster biogenesis, and tRNA processing via RNase P (Chuman & Brody, 1989; Brody *et al*, 1997; Autio *et al*, 2008; Hiltunen *et al*, 2009; Van Vranken *et al*, 2016, 2018). Moreover, mtACP is a key factor in coordinating the cellular response to nutrient availability with intermediary metabolism via its requirement for acetyl-CoA (Hiltunen *et al*, 2010; Kursu *et al*, 2013; Van Vranken *et al*, 2018). If phosphopantetheinylation of mtACP was impeded by a defect in pantothenate kinase 2, then we would expect to see perturbations in these pathways. Specifically, failure to phosphopantetheinylate mtACP would be predicted to lead to impaired complex I activity resulting in decreased oxidative phosphorylation and loss of mitochondrial membrane potential, impaired lipoic acid production with loss of activity of lipoylated enzymes, and impaired iron–sulfur cluster biogenesis with loss of activity of Fe-S-dependent enzymes and processes ultimately leading to iron dyshomeostasis.

Indeed, we observe precisely these changes. mtACP is the NDUFAB1 subunit of complex I, and complex I activity is diminished in our PKAN models (Figs 2E and 4D). Mitochondrial fatty acid synthesis requires the phosphopantetheinyl arm of mtACP to transport reaction intermediates between catalytic components in order to generate acyl-ACPs including octanoyl-ACP, which is the precursor of lipoic acid. Lipoic acid is consumed in the post-translational lipoylation of proteins, including pyruvate dehydrogenase, which requires this modification for its activity. Thus, the decrease in PDH activity that we report (Fig 2D and F) is hypothesized to arise from a decrease in lipoylation resulting from inactive mtACP. Moreover, Lambrechts *et al* (2019) show that under conditions of PANK2 depletion, levels of holo-mtACP were decreased in mammalian cells and in *Drosophila* cells, and lipoylation of PDH was also decreased. Acyl-ACPs are also required for electron transport chain complex assembly and for iron-sulfur cluster biogenesis through their interactions with LYR proteins (Maio *et al*, 2014; Maio & Rouault, 2015). Therefore, loss of acyl-ACPs is the proposed mechanism leading to impaired iron–sulfur cluster formation, which would cause loss of activities of iron–sulfur cluster-dependent complexes and enzymes, including complex I and mitochondrial aconitase, and iron dyshomeostasis. Finally, mtACP is required for processing of mitochondrial tRNAs by RNase P (Autio *et al*, 2008), and a defect in mtACP might further impair respiration by limiting the synthesis of mitochondrially encoded subunits to form other complexes required for electron transport. Though alternative explanations could be proposed for each result that we observe, a defect in mtACP activation explains all of the observed perturbations via a single direct link to CoA, thereby reflecting the most parsimonious interpretation of our data and providing a compelling hypothesis of PKAN pathogenesis.

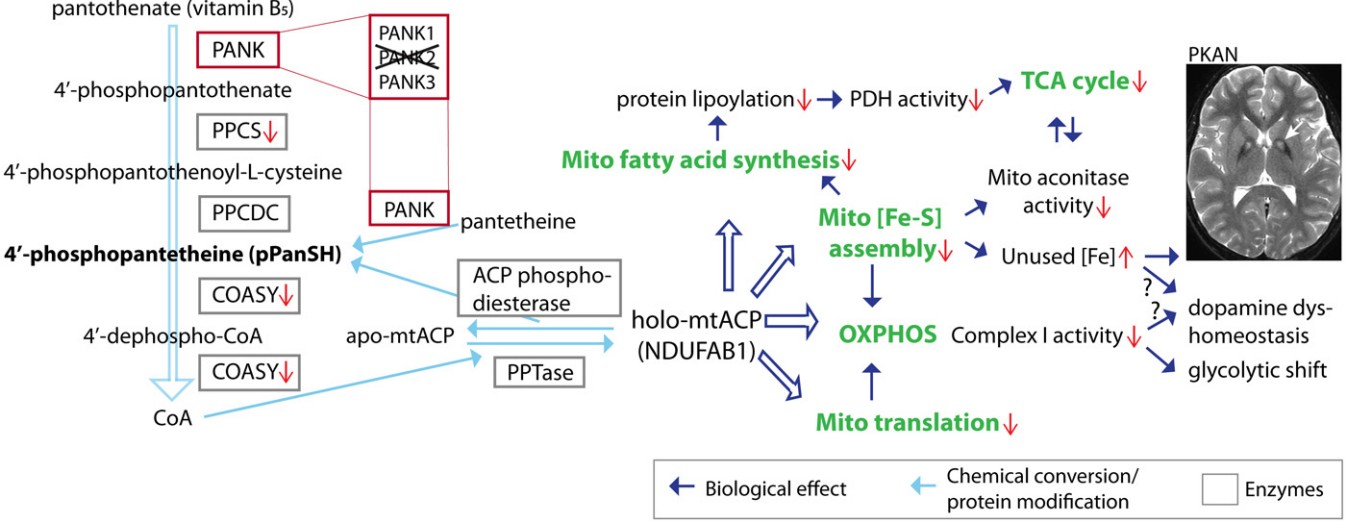

**Figure 8. PKAN pathogenesis is hypothesized to center around mitochondrial ACP and its activation by 4′-phosphopantetheine.**

Our current model predicts that defects in pantothenate kinase 2 impair the phosphopantetheinylation of mtACP and lead to impaired complex I activity (Fig 2E), iron–sulfur biogenesis (Fig 2C), and mitochondrial fatty acid synthesis. Lipoic acid modification is required at the E2 domain of PDH complex. Decreased production of lipoic acid in mitochondria would diminish activity of this enzyme (Fig 2D). The clinical features of PKAN, including abnormal iron accumulation and parkinsonism, can be explained by these metabolic defects. Their rescue by exogenous 4′-phosphopantetheine suggests that the CoA pool critical for mtACP phosphopantetheinylation is at least partially dependent on pantothenate kinase 2.

Why might mtACP function be impaired but not that of other CoA-dependent processes? Lambrechts et al (2019) have indications that levels of CoA are not decreased but that levels of holo-mtACP are decreased. Perhaps differences in the synthesis and regulation of subcellular compartmental CoA pools explain this. We hypothesize that a specific pool of CoA is needed to phospho-pantetheinylate mtACP, that its genesis requires pantothenate kinase 2, that this pool is depleted in PKAN and underlies its molecular pathogenesis, and that exogenous 4′-phosphopantetheine can replenish that pool. However, within the whole cell pools of CoA, this deficiency cannot be detected. The localization of several CoA synthesis enzymes to mitochondria, including PANK2, PPCS, and COASY but not PPCDC (Hortnagel et al, 2003; Johnson et al, 2004; Kotzbauer et al, 2005; Uhlen et al, 2010; Rhee et al, 2013; Dusi et al, 2014, and Appendix Fig S2F), raises the intriguing possibility that CoA pathway intermediates such as 4′-phosphopantetheine might traverse organellar membranes and serve to replenish compartmental pools of CoA (Srinivasan et al, 2015; Sibon & Strauss, 2016). Alternatively, since pantothenate kinases can directly phosphorylate pantetheine to yield 4′-phosphopantetheine (Levintow & Novelli, 1954; Strauss, 2010), mitochondria may depend on pantothenate kinase 2 and this alternate synthesis pathway in order to directly produce CoA destined specifically for the phosphopantetheinylation of mtACP. Although the only known phosphopantetheinyl transferase in mammals localizes to cytoplasm (Beld et al, 2014), a separate mitochondrial phosphopantetheinyl transferase similar to one found in yeast may exist (Stuible et al, 1998). CoA is the required substrate for all phosphopantetheinyl transferases characterized to date; none can use 4′-phospho-pantetheine (Beld et al, 2014). Recent reports that cellular CoA can be synthesized directly from exogenous sources of 4′-phosphopan-tetheine would predict the potential for this key intermediate to

rescue cellular changes in our disease models (Srinivasan et al, 2015; Lambrechts et al, 2019). Here, we suggest that exogenous 4′-phosphopantetheine can serve to replenish the CoA pool that is specifically depleted in PKAN.

We show that 4′-phosphopantetheine can correct the primary CoA metabolic defect as well as the secondary changes observed. The brain abnormalities are fully corrected at the level of gene expression, protein quantity, and enzyme activity following the oral administration of 4′-phosphopantetheine. These changes in gene expression represent a pharmacodynamic biomarker set that can serve for the preclinical development of PKAN therapeutics. CoA synthase gene expression was also found to be decreased in circulating cells, representing a marker in accessible tissue that could be tracked during an interventional study. Moreover, circulating cell biomarker levels appear to change in parallel with those in brain in our mouse model, supporting their use as a possible pharmacodynamic surrogate marker in clinical trials of rational therapeutics for PKAN.

Only 4′-phosphopantetheine corrected *Coasy* expression in our model; vitamin B5, pantetheine, and CoA failed to normalize this biomarker. Neither pantetheine nor CoA is stable in serum. CoA is probably dephosphorylated by intestinal phosphatases and subsequently catabolized to pantetheine, and pantetheine is degraded by serum pantetheinases to pantothenate and cysteamine (Shibata et al, 1983; Wittwer et al, 1985). Thus, neither of these compounds when fed orally would be expected to reach the target tissue intact. In contrast, 4′-phosphopantetheine seems to escape these degradations. The fact that oral administration of 4′-phosphopantetheine normalizes brain biomarkers in our *in vivo* model strongly suggests that this molecule is not degraded by intestinal phosphatases, readily crosses membranes, and reaches brain intact. This is further supported by published data documenting measurable quantities of

endogenous 4′-phosphopantetheine in wild-type mouse serum (Srinivasan *et al*, 2015).

Our disease model studies have revealed new information about the CoA pathway and its regulation. Loss of pantothenate kinase 2 function diminishes levels of *Ppcs* and *Coasy* expression to ~40–60% of control levels. This difference is hypothesized to reflect the portion of CoA synthesis that is specifically attributable to pantothenate kinase 2. We further suggest that regulation of expression of the CoA synthase gene can be controlled by levels of substrate, 4′-phosphopantetheine, and product, CoA, either directly or possibly via the CoAlation of transcription factors or regulatory proteins (Tsuchiya *et al*, 2017, 2018; Gout, 2018).

Important questions have been raised by our studies that remain unanswered. What is the basis for the selective vulnerability of GP to a loss of function of pantothenate kinase 2? There are myriad possibilities but no data yet to support any one; they include regional differences in energy demand, cell membrane composition, mitochondrial fatty acid synthesis, iron–sulfur cluster biogenesis, CoA demand, vascular anatomy, and vulnerability to hypoxia. PKAN biochemical changes are regionally specific in brain; so, why are similar abnormalities seen in "non-diseased" tissues such as fresh lymphocytes and cultured fibroblasts? For this observation, we have no strong hypothesis. Given the apparent capacity for exogenous 4′-phosphopantetheine to easily traverse cell membranes, why are endogenous sources from other subcellular compartments, cells, or tissues unable to mitigate the disease process in PKAN? Our incomplete knowledge about the CoA pathway and organellar differences in CoA metabolism and regulation precludes our offering a compelling hypothesis at this time.

Brain iron accumulation is a prominent feature of PKAN, yet the mechanism underlying iron dyshomeostasis has perplexed the field since discovery of the pantothenate kinase 2 gene (Zhou *et al*, 2001). We show that cells in GP are correctly sensing and responding to high iron in both the cytosolic and mitochondrial fractions. However, we also show impaired function of an iron–sulfur cluster-dependent enzyme, mitochondrial aconitase, suggesting that these cells are starved for bioavailable iron. Impaired mitochondrial function, insufficient iron--sulfur cluster biogenesis, or sequestration of iron could all lead to errant signaling between cytosol and mitochondria to acquire more iron. Loss or mis-sensing of bioavailable iron is a common theme leading to mis-communication between organelles and signaling for increased uptake, a phenomenon that is hypothesized to explain iron accumulation in many neurodegenerative disorders (Babcock *et al*, 1997; LaVaute *et al*, 2001; Calmels *et al*, 2009; Richardson *et al*, 2010; Rouault, 2012; Matak *et al*, 2016). Our data suggest that this mechanism is at work in PKAN, as well.

This concept calls into question the rationale for chelation as a therapeutic approach in PKAN and other neurodegenerative disorders if bioavailable iron is, in fact, insufficient. Alternatively, chelation may complement a rational therapeutic approach by accelerating the removal of mis-sequestered iron that might be contributing to disease (Klopstock *et al*, 2019). We note that exogenous 4′-phosphopantetheine corrects iron dyshomeostasis in both our *in vivo* and *in vitro* models, suggesting that resolution of the primary CoA defect may be sufficient to correct downstream sequelae, including iron accumulation. Though we have not proven that correction of the iron defect derives directly from correction of the

CoA metabolic pathway defect, this would be the most parsimonious interpretation of our data. Moreover, this interpretation would fit with the prediction that re-activation of mtACP would correct iron–sulfur biogenesis. On this basis, we suggest that rescue of the CoA metabolic defect in PKAN may be sufficient to enable cells to re-establish iron homeostasis without any other interventions. While loss of iron–sulfur clusters is a compelling explanation for the observed decrease in mitochondrial aconitase activity, this defect may instead result from slowing of the tricarboxylic acid cycle due to lack of CoA or from lack of acetyl-CoA due to a defect in PDH. To investigate such mechanistic questions, these tractable disease model systems now can serve as critical tools.

Our findings of a dopamine metabolic defect are consistent with the clinical features of dystonia and parkinsonism that are observed in PKAN. Dopaminergic neurons are known to be vulnerable to complex I dysfunction resulting from exposure to the selective toxins rotenone and MPTP (Ferrante *et al*, 1997; Sriram *et al*, 1997; Greenamyre *et al*, 1999; Matak *et al*, 2016). While the mechanism for this vulnerability is unclear (Choi *et al*, 2008), we hypothesize that a complex I defect, possibly in combination with oxidative damage from increased iron, underlies the dopaminergic defect in PKAN. Furthermore, iron is a cofactor for tyrosine hydroxylase activity, leaving open the possibility that iron dyshomeostasis might contribute in multiple ways to the dopamine metabolic perturbations that we see in mouse brain. We note that the dopamine defect is reversible and propose that this results from correction of the CoA pathway defect and subsequent rescue of complex I function and iron homeostasis.

Taken together, our results provide strong evidence for a pharmacodynamic biomarker set that includes proximal, remote, and surrogate markers for use in preclinical as well as clinical studies. These markers of disease are linked to a cascade of molecular changes that can explain key features of PKAN, including pallidal iron accumulation. These features are recapitulated in our mouse disease model, in which all perturbations can be specifically attributed to a defect in pantothenate kinase 2, an important fact that distinguishes this from other published models. We have identified new factors that regulate the CoA pathway. In addition, we provide a new framework and research resources with which to investigate the cellular role of pantothenate kinase 2 and the vulnerability of the globus pallidus to its dysfunction. Finally, we offer strong evidence in support of a CoA pathway intermediate, 4′-phosphopantetheine, as a therapeutic for PKAN.

## Materials and Methods

### Animals and Cells

A murine model with germline *Pank2* null mutation on C57/BL6 background was generated by inserting a stop codon into exon 2 as reported previously (Kuo et al, 2005). This background was maintained by backcrossing to pure C57/BL6 mice (Jackson Laboratories) every 3–4 years. Mice were housed at Oregon Health & Science University vivarium under the care of Department of Comparative Medicine. Animals were housed under 12-h light cycle and cared with daily monitoring while having free access to chow and water. All experimental protocols were pre-approved by

OHSU IACUC (protocol no; IP00000450) and following NIH Office of Laboratory Animal Welfare and NC3Rs guidelines. All animals used for experiments were between 3–6 months of age and $n = 3$–11 per group in each experiment, and both genders were used. The exact number of animals used in each experiment is listed in the figure legend.

Human primary fibroblasts and lymphoblasts were generated using biopsied samples from PKAN patients and age-matched healthy controls using previously published methods (Rittie & Fisher, 2005; Darlington, 2006). Briefly for fibroblasts, following informed consent, a 2-mm skin punch biopsy was obtained with local anesthesia and plated after digesting with proteinases at 37°C. After 2–3 weeks, the tissue piece was removed from culture, and fibroblasts were expanded and stored at the lowest passage number. For the lymphoblasts, leukocytes were separated from patient blood using gradient centrifugation (Histopaque, Millipore Sigma) and transformed using Epstein–Barr virus (ATCC). This transformation of human blood cells is approved by OHSU Institutional Biosafety Committee (protocol no. IBC-11-28). All subjects and the sample collection for the repository are part of OHSU's IRB-approved protocol e7232 with additional work covered by protocol e144.

### Human sample collection using PAXgene blood RNA tubes

Informed consent was obtained from all subjects, and the experiments conformed to the principles set out in the WMA Declaration of Helsinki and the Department of Health and Human Services Belmont Report. All subjects were consented as part of OHSU's IRB-approved protocol e7232 with additional work covered by protocol e144. Human blood from PKAN patients and age-matched controls was collected using PAXgene® Blood RNA tubes (BD Biosciences). After a 24-h incubation to ensure lysis of red blood cells, remaining cells were pelleted and stored at −80°C. Later, total RNA was isolated (Qiagen) and analyzed using qRT–PCR method.

### Mouse brain trisection

Mouse brain was removed from skull and cut sagittally in the middle. The olfactory bulb was removed, and the rest of the hemisphere was cut at Bregma 0.25, −2 and −5 mm. Cortex was removed from the GP-containing piece to generate a GP-enriched sample (GP). Cortex and superior colliculus were removed from the SN-containing piece to yield a SN-enriched sample (SN). Brainstem was removed from the posterior brain piece to isolate cerebellum. There were no differences in trisected area between genotypes in terms of tissue atrophy, tissue weight, or total protein per tissue weight (data not shown). This was a relatively crude dissection method for each area, and other brain areas were present in the GP- and SN-enriched samples. These areas are thalamus, hypothalamus, striatum and some parts of pallidum for GP, and part of the midbrain including ventral tegmental area, red nucleus, and oculomotor nucleus for SN. Enrichment of each area was confirmed by qRT–PCR of known region-specific genes (Appendix Fig S1B).

### qRT–PCR

Total RNA was isolated from various samples, and genomic DNA was removed using gDNA eliminator column (Qiagen). Total RNA

from both mouse brain and PAXgene Blood RNA tube-isolated lymphocytes was extracted using phenol/chloroform method with QIAzol (Qiagen), and primary human cell RNA was extracted using RNeasy Plus Mini Kit (Qiagen). Total RNA concentration was measured using Epoch plate reader (BioTek), and 1 μg of RNA was subject to reverse transcription following manufacturer's protocol (SuperScript™ III First-Strand Synthesis System, ThermoFisher Scientific). Real-time analyses of these cDNA were performed using a Rotor-Gene Q real-time thermal cycler and Rotor-Gene SYBR® Green PCR Kit (Qiagen) following manufacturer's protocol. All data were normalized to the expression of housekeeping genes (*Gapdh* for mice, *18s* for human) and then normalized to the control group expression (Comparative $C_T$ Method, $\Delta\Delta C_T$). Two factors that are crucial for the $\Delta\Delta C_T$ method, the equal expression of housekeeping gene and primer efficiencies, were listed in Appendix Fig S1A. The complete list of primers can be found in Appendix Table S1.

### Westerns and antibodies

Total protein was isolated and quantified using a modified Bradford assay (Bio-Rad). Western blot analyses were performed using the following primary antibodies: mouse anti-COASY (Santa Cruz sc-393812, 1/1,000), rat anti-D1 dopamine receptor (Sigma D2944, 1/1,000), mouse anti-TFR1 (ThermoFisher 13-6800, 1/3,000), rabbit anti-pyruvate dehydrogenase (Cell Signaling 3205, 1/3,000), rabbit anti-VDAC (Cell Signaling 4661, 1/5,000), mouse anti-MSH6 (BD 610918, 1/10,000), and mouse anti-β actin (ThermoFisher AM4302, 1/10,000). Most of the primary antibody binding was visualized with the traditional method using a HRP-conjugated secondary antibody (Jackson ImmunoResearch, 1/10,000) and ECL substrate (SuperSignal™ West Pico PLUS, ThermoFisher). However, anti-DRD1 and its loading control (anti-MSH6) were visualized using a biotinylated secondary antibody and Alexa Fluor™ 488-conjugated streptavidin (ThermoFisher, 1/4,000). The fluorescent Western blot signal was captured using iBright™ FL1000 (ThermoFisher).

### Samples, densitometry, and statistical analysis

For human studies, sample size was determined based on the number of available human patients and their age-matched controls. For mouse studies, minimum of three animals per genotype and treatment group to maximum of 11 were used to ensure statistical analyses. All samples were assigned with non-descriptive, anonymizing ID to hide genotype and treatment status. Samples were equally distributed into various treatment groups. All biochemical analyses that resulted in bands were measured using Image Studio™ Lite by Li-Cor (www.licor.com/bio/image-studio-lite). Statistical analyses were performed using SigmaStat 4.0 (Systat Software) with either one-way ANOVA or two-way ANOVA depending on the number of variables within experiments. All the dataset passed the Shapiro–Wilk normality test and Brown–Forsythe equal variance test (SigmaStat). Statistical significance was determined when $P < 0.05$ and labeled with * or **$P < 0.001$, or ***$P < 0.0001$ in figures to avoid clutter. Most of the data figures contain only biologically meaningful statistical analyses, with the full analyses provided throughout the Appendix figure section and Appendix Table S2. Unless stated specifically in the figure legend, all the data were presented as mean ± s.e.m.

## ICP-MS including mitochondrial isolation

Dissected mouse brain was snap-frozen in liquid nitrogen and stored in a nitric acid-washed tube at −80°C. Crude mitochondrial isolation was performed using a Mitochondria Isolation Kit (Abcam) and checked for cross-contamination in each fraction using Western blot assays. Both mitochondrial and cytosolic fractions were analyzed for various elements in collaboration with Dr. Martina Ralle (Elemental Analysis Core, OHSU) using ICP-MS (Inductively Coupled Plasma Mass Spectroscopy). A small portion of each subcellular fraction was used to measure DNA quantity (Qiagen) and used as the normalization factor.

## Activity assays

Activities of two [Fe-S] containing enzymes, mitochondrial and cytosolic aconitase, were measured as an in-gel assay as previously published (Tong & Rouault, 2006). Briefly, total protein from mouse brain and liver was extracted and run in a non-denaturing gel and incubated with a reaction buffer containing *cis*-aconitate. The enzymatic reaction produces four distinct bands (two for mitochondrial and two for cytosolic), and the band densities were measured. A duplicated denaturing gel was produced and stained with Coomassie Brilliant Blue to serve as a loading control.

Activities of two mitochondrial enzyme complexes (complex I and PDH) were measured using Dipstick assays following manufacturer's protocols (Ambion). Briefly, total protein from dissected mouse GP or cultured cells was extracted using the extraction buffer with mild detergent, and total protein concentration was measured using Bradford assay. To check potential saturation of capturing antibodies, only WT GP samples were analyzed using increasing amounts of protein. Once a saturation curve was established, approximately ¾ point within the linear range was selected as the loading amount (6 μg for PDH activity assay, and 2 μg for the complex I activity assay). Then, samples from each group were wicked through a Dipstick containing a band of capturing antibodies, and wash buffer was applied. The target-specific substrates were then converted to colored bands and their densities analyzed.

## Extracellular flux analysis and JC-1 assay

Primary non-transformed fibroblasts were cultured in standard media and plated in Seahorse XF24 Analyzer plates (Agilent) the day before each experiment (10,000–20,000 cells per well depending on the growth speed). On the day of the experiment, cells were washed and switched to the XF Base Medium one hour before experiment (2 mM glutamine was added for the ECAR assay per manufacturer's recommendation). Both ECAR and OCR were measured following manufacturer's protocol. After the experiment, cells were washed, and live cells were counted as a normalization factor.

The relative population of cells with active mitochondria was measured using a JC-1 dye following manufacturer's protocol (ThermoFisher Scientific). Briefly, both control and PKAN lymphoblasts were washed and incubated with JC-1, and the emission fluorescent intensities at 595 nm (red) and 535 nm (green) were measured. The red/green ratio from the control cells was set at 100%, and the PKAN cells were plotted as a relative percentage.

### The paper explained

**Problem**

PKAN is a rare, high burden movement disorder for which there are no approved disease modifying agents. The lack of a high-fidelity mouse model has impeded understanding of disease pathophysiology and development of rational therapeutics.

**Results**

Using a mouse null mutant of *Pank2*, the mitochondrial isoform of pantothenate kinase, we show a defect in coenzyme A metabolism that is revealed by careful dissection of brain into disease-vulnerable (GP) and disease-protected (SN and cerebellum) regions, an idea based on the exquisitely focal pathology found in the human disorder. In addition, we see increased cellular iron levels and perturbations in iron-related proteins and in iron-sulfur cluster-dependent enzyme activities. Defects in electron transport chain function and pyruvate dehydrogenase activity were also observed. Mitochondrial acyl carrier protein activation by coenzyme A is hypothesized to be the central factor tying together these disparate metabolic changes and underlying the molecular cascade of PKAN pathogenesis. All metabolic perturbations are normalized by oral administration for 2 weeks of a CoA pathway intermediate, 4′-phosphopantetheine.

**Impact**

We describe a mouse model of PKAN in which all features are specifically attributable to loss of pantothenate kinase 2 activity. Recapitulation of key human disease features, including iron accumulation in globus pallidus and dopamine dyshomeostasis, supports the fidelity of this model and its utility for delineating disease pathogenesis and testing candidate therapeutics. Correction of all disease features by 4′-phosphopantetheine provides compelling evidence to support a clinical trial in PKAN.

## Treatment protocols

Both PPARα agonists were purchased from Millipore Sigma. Both WT and KO mice were treated with bezafibrate (0.8 mg/g body weight) or gemfibrozil (1.2 mg/g body weight) by mixing with ground standard mouse chow for 14 days. Singly caged mice had limited access to food to ensure correct dose delivery (170 mg chow/g body weight). After 14 days, mice were sacrificed and GP area was dissected and analyzed for expression of a gene considered to serve as a positive control for treatment (*Cpt1c*, Appendix Fig S5B).

Both pantetheine and vitamin B$_5$ were purchased from Millipore Sigma. Coenzyme A was purchased from CoALA Biosciences (Austin, TX), and highly pure (> 97%) 4′-phosphopantetheine (pPanSH) was provided by Dr. Ody Sibon (University of Groningen, the Netherlands). For the efficacy test (Fig 6A), all drugs were delivered orally in 10% sucrose solution for 14 days at 5 μg/g for pPanSH and at molar equivalent dosages for all others. After 14 days, mice were sacrificed, and dissected brain was analyzed for gene expression recovery. After this experiment, a dose determination experiment was performed using only pPanSH from 0 to 20 μg/g dose range (Fig 6B and C). For the *in vitro* study, primary human fibroblasts were treated with pPanSH at various concentrations (0–200 μM) for 24 h. Cells were then washed and pelleted, and qRT–PCR analyses were performed to analyze expression of various genes. For the extracellular flux analysis, cells were treated

with 50 μM pPanSH for three days in standard culture media, and pPanSH was replenished every 24 h based on its half-life in serum (Srinivasan et al, 2015).

For the withdrawal study, both WT and KO mice were treated with either vehicle or pPanSH at 5 μg/g for 14 days. Each group of mice was then sacrificed at day 0, 1, 2, 3, or 7 and their brains dissected and prepped to analyze the duration of effect of pPanSH.

**Expanded View** for this article is available online.

## Acknowledgements

We are grateful to PKAN families worldwide, who have partnered with us in this important project and provided steady support and inspiration. This work was supported by NIH R21HD088833 (S.J.H.) and R01NS109083 (S.J.H.), the NBIA Disorders Association (S.Y.J.), Friends of Doernbecher (P.H.), the OCTRI Biomedical Innovation Program (P.H.), and the Collins Foundation (P.H.). The OHSU Medicinal Chemistry Core was instrumental in supporting this work through creativity, perseverance, and drive. We thank the OHSU Elemental Analysis Core for expert support. Research reported in this publication was supported by the National Center for Advancing Translational Sciences of the National Institutes of Health under award number UL1TR0002369. The content is solely the responsibility of the authors and does not necessarily represent the official views of the National Institutes of Health.

## Author contributions

SYJ, SJH, OCMS, PH, RL, and RW designed the research studies, analyzed data, and wrote the manuscript. SYJ, RF, JC, TP, NG, MZ, MR, and MD conducted experiments, and acquired and analyzed data. AMG, RF, JH, DZ, LS, PR, AF, and KW acquired samples or conducted experiments. AP, HJ, and AN generated essential reagents.

## Conflict of interest

Drs. Jeong, Hayflick, and Hogarth are co-inventors on a patent application for the PKAN biomarker set. Drs. Hayflick and Sibon are co-inventors on a patent application for 4′-phosphopantetheine for use in disorders exclusive of PKAN. Dr. Sibon is a co-inventor on a patent application for acetyl-4′-phosphopantetheine for use in PKAN and in related disorders. Dr. Hayflick is a non-compensated member of the Scientific Advisory Board of BioPontis Alliance, a non-profit organization. Drs. Hayflick and Hogarth are non-compensated members of the Scientific and Medical Advisory Board of the NBIA Disorders Association, a non-profit lay advocacy organization. Dr. Hayflick is a non-compensated member of the Scientific and Medical Advisory Board of the NBIA Alliance, a non-profit lay advocacy organization. Drs. Hayflick and Hogarth serve as non-compensated executives for the Spoonbill Foundation, a not-for-profit organization that may benefit from the results of this research and technology. This potential conflict of interest has been reviewed and managed by OHSU.

## For more information

(i) http://nbiacure.org

(ii) https://www.nbiadisorders.org

(iii) http://www.nbiaalliance.org

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
