## [Review Process File · EMBO Molecular Medicine]

4'-Phosphopantetheine corrects CoA, iron and dopamine metabolic defects in mammalian models of PKAN

Suh Young Jeong, Penelope Hogarth, Andrew Placzek, Allison M Gregory, Rachel Fox, Dolly Zhen, Jeffrey Hamada, Marianne van der Zwaag, Roald Lambrechts, Haihong Jin, Aaron Nilsen, Jared Cobb, Thao Pham, Nora Gray, Martina Ralle, Megan Duffy, Leila Schwanemann, Puneet Rai, Alison Freed, Katrina Wakeman, Randy Woltjer, Ody C M Sibon, & Susan J Hayflick

Review timeline:

Submission date:	5 March 2019
Editorial Decision:	10 April 2019
Revision received:	10 July 2019
Editorial Decision:	29 July 2019
Revision received:	7 August 2019
Accepted:	14 August 2019

Editor: Céline Carret

Transaction Report:

1st Editorial Decision

10 April 2019

Thank you for the submission of your manuscript to EMBO Molecular Medicine. We have now heard back from the two referees whom we asked to evaluate your manuscript.

You will see from the set of comments below that both referees find the study important, and while ref. 1 is supportive of publication, ref. 2 is more hesitant and raises some pertinent issues: relevance of the mouse model, phosphopantetheine treatment rescue-effect to dissect, as well as sub-cellular localisation of PPCS and PPCDC to assess. We would also like you to fix some technical limitations and provide more details and explanations. Both referees agree that the link between the two papers should be made stronger (same comment went to the other paper).

Upon our cross-commenting exercise, ref. 2 added "As it stands, the only link highlighted by both papers in that the impaired mtACP biogenesis result reported in the Sibon ms [A single experiment (Fig. 2) conducted in S1 insect cells, supported by one of the kd experiments shown in Fig. 4 (E' and F')]. For back-to-back papers, I would have liked to see greater interdependency. In some respects (as also noted by Ref. 1) the two papers currently actually mutually detract, and consequently reduce the overall impact of the other paper." Further this referee also added "The enzyme activity data is not convincing. And the authors do not correctly represent our current knowledge of the subcellular localization of the CoA biosynthetic enzymes."

We would therefore welcome the submission of a revised version within three months for further consideration and would like to encourage you to address all the criticisms raised as suggested to improve conclusiveness and clarity. Please note that EMBO Molecular Medicine strongly supports a single round of revision and that, as acceptance or rejection of the manuscript will depend on another round of review, your responses should be as complete as possible.

I look forward to receiving your revised manuscript.

***** Reviewer's comments *****

Referee #1 (Comments on Novelty/Model System for Author):

The exact mechanism used to generate the mouse ko should be described or referenced. It appears to be a total ko- what was the approach used to make this model

Referee #1 (Remarks for Author):

This paper submission represents the culmination of years of preparatory work and it overcomes previous roadblocks. The dissection of the brain into sections, focusing on GP, proved to very useful. The molecular amelioration of feeding 4 phosphopantetheine to animals is impressively clear. In the next paper, the authors could consider using IH with some of their markers to see if a specific subset of GP cells are affected. It was probably very good for their study that the mice did not develop advanced disease.

Minor

They need to explain in greater detail exactly which mouse model is used here and how it was generated.

Referee #2 (Comments on Novelty/Model System for Author):

One of the central points of the manuscript is the study of the model system to determine its utility/adequacy; in this sense this question is therefore not entirely relevant.

Referee #2 (Remarks for Author):

In the manuscript by Jeong et al. the authors set out to establish an improved mammalian model of PKAN, the neurodegenerative disease associated with a genetic defect in the first enzyme of coenzyme A (CoA) biosynthesis, pantothenate kinase (PanK). This was achieved by isolating brain tissues known to be affected by the disease from healthy tissue and by identifying biomarkers in the isolated tissues that show changes that correlate with disease status, and that can be corrected by intervention with a CoA biosynthetic pathway intermediate that bypasses the PanK defect.

The described research is an excellent observational study rich in data obtained from a variety of samples and different treatment conditions. However, the data-rich results, and specifically the manner in which it is presented unfortunately makes it difficult to determine a) what the main goal of the study was and b) whether the data interpretation is unambiguous, i.e. whether other conclusions would not be equally valid. Consequently, it is not obvious how the outcome should be evaluated. This reduces the overall impact of the manuscript in its current form.

To highlight a few specific concerns:

1. Model organism and biomarker identification: The manuscript's biggest strength is in the studies done to establish an animal model for PKAN, and to identify biomarkers of the disease. The finding that a Pank2 KO mouse that does not show a strong and immediate disease phenotype indeed shows differences in the expression of key genes (determined through qRT-PCR) in the disease-vulnerable globus pallidus (GP) is an important finding. However, while implied by the nature of the analysis, the authors should concede that the utility of the model and the biomarkers may be limited due to the invasive nature of the analysis, which requires sacrifice of the animals. Or they need to be

clearer in how they see the model being used in the study of the disease and treatments - this relates to their use of fibroblasts as surrogate for analysis (See also point 4 below).

2. Biomarker analysis: The bulk of the biomarker analysis is done by qRT-PCR. In this context it is important to provide evidence of the robustness of the method and the associated statistical analysis. Specifically, the authors used the Comparative CT method (ddCT) to perform quantification. This methodology assumes 100% PCR efficiency in all samples; if there are differences in the PCR efficiency between the reference gene and the gene of interest, this method will yield greatly skewed results. The biomarker analysis will be significantly improved if a method is used that takes this variation in efficiency into account, such as the one proposed by Pfaffl (PMID: 11328886). Moreover, the authors chose to report all the data in a set as % values relative to the mRNA level related to the gene of interest in the GP. Are they certain that the reference gene's expression levels are the same across all brain regions? If not, then the comparison cannot be made in this manner. Finally, I'm concerned about the statistical analysis when there is such a large variation in the number of independent samples used in a specific experiment. E.g. in the case of the data presented in Figs. 2a and 3c the value of n varies between 3 and 11. Why such a big difference? Although ANOVA analysis usually takes account of such differences, it would be preferable if the sample sizes were more equally matched. Moreover, for the data in Figure 1, two sets of n values are given but it is not clear what each refers to. Overall, considering that the conclusions of the study stands and falls with the biomarker analyses, I would like to see a more thorough presentation of the method and the statistical treatment of the results.

3. Enzyme activity analysis: The authors report activity assays for aconitase (fig. 2c), PDH in the GP (fig. 2d) and complex I in the GP (Fig. 2e). However, I have several concerns about this data. First, the authors are not clear how the results are normalized to total protein in the sample, or how the sample sizes (i.e. amount of cells used for preparations of the lysates) are kept similar. The relevant statements in the methods section is: "Total protein...[]...was extracted". Was the amount of tissue used the same? How was the protein extracted? For the aconitase assay, how much protein was loaded? Was this standardised? For the PDH assay, can the authors be sure that the Dipstick was not saturated with PDH? The assays as presented here are all in reporting specific activities, which is an indication of the amount of active enzyme relative to the total amount of protein. It is therefore important to be clear on how the total amount of protein was determined and how it was kept comparable between the samples.

4. Correlation of data between systems: The authors indicate that their findings in mouse was corroborated by using primary cells from PKAN patients (fibroblasts and lymphoblasts). Yet, in the previous section the focus was to show a distinction in the expression levels of the biomarkers in the different brain tissues in the mouse. Why do they expect the expression in primary blood cells to correspond with those in the GP? How can they be sure that this correlation relates to the disease phenotype, particularly if they spend so much effort in showing that there is NO correlation between the expression in different brain tissues, and between brain and liver?

5. Effects of phosphopantetheine treatment: The authors demonstrate effects on the biomarker expression levels following phosphopantetheine treatment, but not following treatment with CoA, pantetheine, or Vit B5. However, the authors acknowledge in the discussion that this is not unexpected as the other test compounds are all expected to be broken down. Consequently, the value of such a negative result is rather questionable. In regards to the effect on Coasy expression following phosphopantetheine treatment the authors conclude that high levels repress its expression in both WT and KO animals; yet the levels are similar in the treated and not treated KO. It is therefore unclear if any repression is actually happening, or if the lower levels of phosphopantetheine just alleviates the basal repression seen in the KO animal. Again, as alluded to in the previous point, it is unclear how the results obtained from the human fibroblasts can be related to those from the mouse GP.

6. Hypothesis of PKAN pathogenesis: The authors use the hypothesis proposed by Lambrechts et al (the paper submitted together with theirs) to explain their results. Indeed, in my opinion this is the only reason why the current paper deserves to be combined by the one by Lambrechts; however, it could equally be evaluated and understood separately from it. Be that as it may, a key point not explained by this model, and which the authors fail to address, is why the phosphopantetheine produced by the cytosolic biosynthesis of CoA fails to provide similar rescue as seen when

phosphopantetheine is exogenously applied. Indeed, a recent report indicated that activation of the cytosolic PanK enzymes could alleviate PKAN-related phenotypes (PMID: 30352999). Moreover, the model doesn't explain why the effects are uniquely detrimental to the GP, while the impact on CoA, mtACP and all corresponding factors should be the same in all cells. The key question here is subcellular localization of the PPCS and PPCDC enzymes, and the permeability of the mitochondrial membrane to CoA intermediates - and whether this differs by cell type/tissue. Although the authors state in the introduction that CoA biosynthesis is cytosolic, to my knowledge no evidence has been published on the subcellular localization of the PPCS and PPCDC enzymes. Since these enzymes are able to form phosphopantetheine from the product of the PanK reaction, having this knowledge is key to the complete understanding of PKAN pathogenesis, and treatment options.

Apart from addressing the questions above, the authors should also pay attention to the overall writing and structure of the manuscript. For the reader not familiar with the disease or the genes in question, it would be helpful to have a bit more context provided when these are first mentioned (eg. the genes referred to in the first paragraph on p. 8). There also seem to be an inconsistent handling of the gene and protein nomenclature in regards to capitals/italics in the text and figures/figure legends (note specifically the various forms of Coasy). Finally, some of the figure legends are not completely clear without reference to the text.

Taken together, I believe that it would be premature to publish the manuscript in its current form.

1st Revision - authors' response

10 July 2019

Referee #1 (Comments on Novelty/Model System for Author):

The exact mechanism used to generate the mouse ko should be described or referenced. It appears to be a total ko- what was the approach used to make this model

See below.

Referee #1 (Remarks for Author):

This paper submission represents the culmination of years of preparatory work and it overcomes previous roadblocks. The dissection of the brain into sections, focusing on GP, proved to very useful. The molecular amelioration of feeding 4 phosphopantetheine to animals is impressively clear. In the next paper, the authors could consider using IH with some of their markers to see if a specific subset of GP cells are affected. It was probably very good for their study that the mice did not develop advanced disease.

Minor

They need to explain in greater detail exactly which mouse model is used here and how it was generated.

We appreciate Referee #1's suggestion and will explore cell specific changes in a future manuscript.

We clarified that the mouse model is a whole-body, germline Pank2 null mutant mouse in the 'Main' and 'Methods' sections and referenced the original report.

Referee #2 (Comments on Novelty/Model System for Author):

One of the central points of the manuscript is the study of the model system to determine its utility/adequacy; in this sense this question is therefore not entirely relevant.

Referee #2 (Remarks for Author):

In the manuscript by Jeong et al. the authors set out to establish an improved mammalian model of PKAN, the neurodegenerative disease associated with a genetic defect in the first enzyme of coenzyme A (CoA) biosynthesis, pantothenate kinase (PanK). This was achieved by isolating brain tissues known to be affected by the disease from healthy tissue and by identifying biomarkers in the isolated tissues that show changes that correlate with disease status, and that can be corrected by intervention with a CoA biosynthetic pathway intermediate that bypasses the PanK defect.

The described research is an excellent observational study rich in data obtained from a variety of

samples and different treatment conditions. However, the data-rich results, and specifically the manner in which it is presented unfortunately makes it difficult to determine a) what the main goal of the study was and b) whether the data interpretation is unambiguous, i.e. whether other conclusions would not be equally valid. Consequently, it is not obvious how the outcome should be evaluated. This reduces the overall impact of the manuscript in its current form.

See above General responses.

To highlight a few specific concerns:

1. Model organism and biomarker identification: The manuscript's biggest strength is in the studies done to establish an animal model for PKAN, and to identify biomarkers of the disease. The finding that a Pank2 KO mouse that does not show a strong and immediate disease phenotype indeed shows differences in the expression of key genes (determined through qRT-PCR) in the disease-vulnerable globus pallidus (GP) is an important finding. However, while implied by the nature of the analysis, the authors should concede that the utility of the model and the biomarkers may be limited due to the invasive nature of the analysis, which requires sacrifice of the animals. Or they need to be clearer in how they see the model being used in the study of the disease and treatments - this relates to their use of fibroblasts as surrogate for analysis (See also point 4 below).

We have added more specific information about the utility and limitations of this animal model for studies of disease pathogenesis and potential therapeutics. While the need to sacrifice animals to study specific brain abnormalities does pose limitations, this is a common challenge with mammalian models of neurological disorders.

2. Biomarker analysis: The bulk of the biomarker analysis is done by qRT-PCR. In this context it is important to provide evidence of the robustness of the method and the associated statistical analysis. Specifically, the authors used the Comparative CT method (ddCT) to perform quantification. This methodology assumes 100% PCR efficiency in all samples; if there are differences in the PCR efficiency between the reference gene and the gene of interest, this method will yield greatly skewed results. The biomarker analysis will be significantly improved if a method is used that takes this variation in efficiency into account, such as the one proposed by Pfaffl (PMID: 11328886). Moreover, the authors chose to report all the data in a set as % values relative to the mRNA level related to the gene of interest in the GP. Are they certain that the reference gene's expression levels are the same across all brain regions?

If not, then the comparison cannot be made in this manner. Finally, I'm concerned about the statistical analysis when there is such a large variation in the number of independent samples used in a specific experiment. E.g. in the case of the data presented in Figs. 2a and 3c the value of n varies between 3 and 11. Why such a big difference? Although ANOVA analysis usually takes account of such differences, it would be preferable if the sample sizes were more equally matched. Moreover, for the data in Figure 1, two sets of n values are given but it is not clear what each refers to. Overall, considering that the conclusions of the study stands and falls with the biomarker analyses, I would like to see a more thorough presentation of the method and the statistical treatment of the results.

We appreciate the Referee's concern regarding the primer efficiency and the equal expression of Gapdh as a housekeeping gene, which are important factors that can change the overall results. Therefore, we re-tested all the primer efficiencies used in study using serially diluted cDNA samples. All primer efficiencies were between 90-110%, which is expected for a good SYBR green QPCR reaction.

Target	R ² value	PCR Efficiency (%)
Coasy	0.9995	101.6
Drd1	0.9951	110.7
Gapdh	0.9986	103.1
Ireb2	0.9991	101.0
Pank1α	0.9961	100.3
Ppcs	0.9989	104.1
Tfrc	0.9976	102.8

We also confirmed that expression of the housekeeping gene (*Gapdh*) is not skewed between tri-sected areas. [Unpublished data removed at the authors' request.] Moreover, mRNA expression data from the Allen Brain Atlas also shows relatively equal expression of *Gapdh* throughout mouse brain [unpublished data removed at the authors' request]. Finally, we have clarified number of samples used for each experiment.

3. Enzyme activity analysis: The authors report activity assays for aconitase (fig. 2c), PDH in the GP (fig. 2d) and complex I in the GP (Fig. 2e). However, I have several concerns about this data. First, the authors are not clear how the results are normalized to total protein in the sample, or how the sample sizes (i.e. amount of cells used for preparations of the lysates) are kept similar. The relevant statements in the methods section is: "Total protein...[]...was extracted". Was the amount of tissue used the same? How was the protein extracted? For the aconitase assay, how much protein was loaded? Was this standardised? For the PDH assay, can the authors be sure that the Dipstick was not saturated with PDH? The assays as presented here are all in reporting specific activities, which is an indication of the amount of active enzyme relative to the total amount of protein. It is therefore important to be clear on how the total amount of protein was determined and how it was kept comparable between the samples.

We agree with the Referee that the PDH and complex I activity assays have potential limitations of saturation of the 'capturing antibody'. Therefore, we ran control experiments first in order to examine this issue. Below is the PDH activity dipstick assay using only WT GP samples with increasing amount of protein. Based on these data, we used 6ug of total protein for each group. The manufacturer recommends using ~ 3/4 of total protein within the linear range. The same control experiments were performed for the complex I activity dipstick assay, and 2ug of total protein was used for this assay. [Unpublished data removed at the authors' request.

As for the normalization, we used the same amount of total protein extracted from mouse brain and human cells. This could have posed a problem if there were atrophy in mouse brain or significant differences in cell size between genotypes. However, we have observed no protein synthesis problem or differences in cell size of any samples. As a matter of fact, total protein yield per wet tissue weight (see table below) did not differ between WT and KO mouse GP. We affirm that this information was not reported in detail in the Methods section and have now modified the manuscript accordingly.]

Genotype	GP total protein / wet tissue weight (mg/mg)	Student's t test between genotypes
WT	9.06 \pm 0.84	
KO	8.46 \pm 0.51	0.07

4. Correlation of data between systems: The authors indicate that their findings in mouse was corroborated by using primary cells from PKAN patients (fibroblasts and lymphoblasts). Yet, in the previous section the focus was to show a distinction in the expression levels of the biomarkers in the different brain tissues in the mouse. Why do they expect the expression in primary blood cells to correspond with those in the GP? How can they be sure that this correlation relates to the disease phenotype, particularly if they spend so much effort in showing that there is NO correlation between

the expression in different brain tissues, and between brain and liver?

We were surprised to find the same patterns of biomarker changes in fresh blood cells and cultured fibroblasts, and we lack a strong hypothesis to explain this. Nevertheless, we did observe similar differences in biomarkers in blood and brain at baseline and in response to phosphopantetheine treatment in mice. We do not, however, expect the blood biomarkers to be directly influenced by levels of the brain biomarkers. Instead, we view the blood biomarkers as quantifiable abnormalities that can be used to determine if oral treatment reaches the circulation, is taken up by cells, and alters gene expression (COASY) or protein function (complex I) in an accessible cell type. We hypothesize that a candidate therapeutic that is delivered systemically to humans and crosses the blood-brain barrier would change expression in blood cells in parallel to changing expression in neurons. In this way, the blood biomarkers may be able to serve as a surrogate for brain changes in human studies. We have reviewed the text to ensure that this idea is clearly presented and appreciate further comments if not.

5. Effects of phosphopantetheine treatment: The authors demonstrate effects on the biomarker expression levels following phosphopantetheine treatment, but not following treatment with CoA, pantetheine, or Vit B5. However, the authors acknowledge in the discussion that this is not unexpected as the other test compounds are all expected to be broken down. Consequently, the value of such a negative result is rather questionable.

Readers who are not immersed in the CoA field may wonder about the effects of administering these related compounds. Moreover, in published cell culture systems, CoA has had some therapeutic effect. Therefore, we propose to retain Fig 6a within the larger figure. At the Editor's request, we could move Fig 6a to supplemental data.

In regards to the effect on Coasy expression following phosphopantetheine treatment the authors conclude that high levels repress its expression in both WT and KO animals; yet the levels are similar in the treated and not treated KO. It is therefore unclear if any repression is actually happening, or if the lower levels of phosphopantetheine just alleviates the basal repression seen in the KO animal.

We acknowledge that this is an alternate explanation of the KO data; however, the observation of a similar decrease in Coasy expression with higher doses of 4'-phosphopantetheine in the WT animals would then require a second explanation. Therefore, we propose the most parsimonious interpretation of our data relying on a single mechanism for decreased expression at higher doses in both genotypes. That said, we have introduced the Referee's concept regarding these results, and we appreciate the alternate idea.

Again, as alluded to in the previous point, it is unclear how the results obtained from the human fibroblasts can be related to those from the mouse GP.

See 4 above.

6. Hypothesis of PKAN pathogenesis: The authors use the hypothesis proposed by Lambrechts et al (the paper submitted together with theirs) to explain their results. Indeed, in my opinion this is the only reason why the current paper deserves to be combined by the one by Lambrechts; however, it could equally be evaluated and understood separately from it.

We agree that their papers could stand alone; however, our view is that Jeong amplifies the medical relevance of Lambrechts, and Lambrechts provides strong experimental data implicating mtACP in support of hypothesis presented in Jeong. Therefore, we prefer to keep the papers together.

Be that as it may, a key point not explained by this model, and which the authors fail to address, is why the phosphopantetheine produced by the cytosolic biosynthesis of CoA fails to provide similar rescue as seen when phosphopantetheine is exogenously applied.

We acknowledge this as a limitation of our current knowledge of the pathway and have now stated that explicitly.

Indeed, a recent report indicated that activation of the cytosolic PanK enzymes could alleviate PKAN-related phenotypes (PMID: 30352999).

This comment raises an important point that we have addressed directly in our paper. The model reported in the referenced paper is not a PKAN-related phenotype; it requires a double-hit to cause global impairment, and the phenotype is no more PKAN-like than is that of any other gait-

impaired mouse. As an example, cerebellar loss of Pank1+Pank2 may cause ataxia and gait abnormalities, one of many feasible explanations that were not explored. While those authors represent that mouse as PKAN-like, their conclusion is not supported; they created a neuronal CoA-depleted mouse and treated it with a CoA-replenishing compound. Such a model may or may not have relevance to PKAN, as we state in our paper.

Moreover, the model doesn't explain why the effects are uniquely detrimental to the GP, while the impact on CoA, mtACP and all corresponding factors should be the same in all cells.

We acknowledge this as a limitation of our current knowledge of the pathway and have now stated that explicitly. We elected not to present the myriad possibilities that distinguish GP from other regions, including differences in energy demands, differences in cell membrane composition, differences in vascular anatomy, and vulnerability to hypoxia. We have added this list to the discussion around 'why GP?'

The key question [ONE key question] here is subcellular localization of the PPCS and PPCDC enzymes, and the permeability of the mitochondrial membrane to CoA intermediates - and whether this differs by cell type/tissue.

This is an important point, and we appreciate its being raised. We performed new experiments and evaluated public databases to support our conclusion that PPCS is non-exclusively found in mitochondria and PPCDC is in cytoplasm. Moreover, we analyzed their subcellular localization in each brain region and found no differences nor any significant regional variation in their quantities. The Referee raises one possible explanation, which our mouse data now refute, but there are numerous other possible explanations, including differences in demand for mFAS, Fe-S cluster biogenesis or CoA demand, etc. We have corrected our statements about subcellular localization of the PPCS and PPCDC enzymes and have added to the discussion about possible bases for the observed regional differences as noted above.

Although the authors state in the introduction that CoA biosynthesis is cytosolic, to my knowledge no evidence has been published on the subcellular localization of the PPCS and PPCDC enzymes. Since these enzymes are able to form phosphopantetheine from the product of the PanK reaction, having this knowledge is key to the complete understanding of PKAN pathogenesis, and treatment options.

Based on this insightful comment about alternate substrates for pantothenate kinase (specifically pantetheine), we added this idea to our discussion and referenced the published evidence. We thank the Referee.

Apart from addressing the questions above, the authors should also pay attention to the overall writing and structure of the manuscript. For the reader not familiar with the disease or the genes in question, it would be helpful to have a bit more context provided when these are first mentioned (eg. the genes referred to in the first paragraph on p. 8). There also seem to be an inconsistent handling of the gene and protein nomenclature in regards to capitals/italics in the text and figures/figure legends (note specifically the various forms of Coasy). Finally, some of the figure legends are not completely clear without reference to the text.

We have edited the manuscript to introduce genes and proteins/enzymes early in the paper. In reference to mouse/human and gene/protein nomenclature, we have strived to accurately format these to reflect which we are talking about at every point in the manuscript. If questions remain, we appreciate our being pointed to specific questions of nomenclature. The legends have been modified to enable them to be understood without reference to the text.

2nd Editorial Decision

29 July 2019

Thank you for the submission of your revised manuscript to EMBO Molecular Medicine. We have now received the enclosed report from the referee who was asked to re-assess it. As you will see the reviewer is now globally supportive and I am pleased to inform you that we will be able to accept your manuscript pending minor editorial amendments and a response addressing the minor changes commented by referee 2.

Please submit your revised manuscript within two to three weeks. I look forward to seeing a revised form of your manuscript as soon as possible.

***** Reviewer's comments *****

Referee #2 (Remarks for Author):

The authors clearly attempted to address all the concerns raised in an appropriate manner. This has raised the quality and presentation of the manuscript to a level where publication would be warranted following appropriate revision of the following points:

- 1) The authors carefully responded to the questions raised regarding the qRT-PCR, but I do not see this information included in the manuscript, only in the rebuttal. I would suggest that it is included in the supplementary materials to support the quality of the analysis.
- 2) As requested, the authors have now clarified the number of samples used for the various experiments. However, the authors did not indicate whether they are concerned about the influence of the large variance in sample sizes (in some cases) on the statistical analysis. I'm not a statistician, and I will not push this point; yet in the context of validating biomarkers I would want to be certain that the data analysis is not skewed because of the use of unequal samples sizes.
- 3) I do not see the information that total protein yield per wet tissue weight was the same for WT and KO mouse GP included in the manuscript/SM. This is useful and should be added (perhaps as a statement in the methods?)
- 4) In regards to the localization of PPCS/PPCDC, the authors state in the rebuttal: "We performed new experiments and evaluated public databases to support our conclusion that PPCS is non-exclusively found in mitochondria and PPCDC is in cytoplasm. Moreover, we analyzed their subcellular localization in each brain region and found no differences nor any significant regional variation in their quantities."

However, apart from the following statement found in the discussion:

"The non-exclusive mitochondrial localization of several CoA synthesis enzymes, including PANK2, PPCS, and COASY, raises the intriguing possibility that CoA pathway intermediates such as 4'-phosphopantetheine might traverse organellar membranes and serve to replenish compartmental pools of CoA"

I do not see anything in the revised manuscript that reflects the outcome of these "new experiments" and analysis. The authors do provide references associated with this statement, but since they have gone through the trouble of doing the analysis I want to suggest that they are explicit with the reasons/basis for the statement.

In addition, I find the use of "non-exclusive mitochondrial localization" in the quoted statement somewhat problematic. First, this phrase also refers to PANK2, which is stated to be **UNIQUELY** localized in the mitochondria in the introduction. Using "uniquely" in the first instance and then "non-exclusively" here seems contradictory; if it isn't, the authors should be clear in what they mean so that there is no confusion.

Second, either PPCS or PPCDC *must* be exclusively localized in one of the subcellular compartments in order to give credence to the idea that there are compartmental pools of CoA. Do the authors propose that PPCDC is not found in the mitochondria? If so, they again must be explicit in this regard, and explain their reasoning for this proposal.

5) Gene/Protein nomenclature: The possible confusion here is in the reference of mouse vs human genes/proteins. On p.18, in the line "With higher doses of 4'-phosphopantetheine, we observed progressively lower levels of Coasy expression in GP" I believe Coasy should be in italics.

6) "phosphopantothenoylcysteine" is spelled without a hyphen.

2nd Revision - authors' response

7 August 2019

***** Reviewer's comments *****

Referee #2 (Remarks for Author):

The authors clearly attempted to address all the concerns raised in an appropriate manner. This has raised the quality and presentation of the manuscript to a level where publication would be warranted following appropriate revision of the following points:

1) The authors carefully responded to the questions raised regarding the qRT-PCR, but I do not see this information included in the manuscript, only in the rebuttal. I would suggest that it is included in the supplementary materials to support the quality of the analysis.

We appreciate the comment and have included these data in Appendix Figure S1A

2) As requested, the authors have now clarified the number of samples used for the various experiments. However, the authors did not indicate whether they are concerned about the influence of the large variance in sample sizes (in some cases) on the statistical analysis. I'm not a statistician, and I will not push this point; yet in the context of validating biomarkers I would want to be certain that the data analysis is not skewed because of the use of unequal samples sizes.

As Reviewer #2 noted in the previous review, ANOVA analysis factors the number of samples as a part of the analysis. Moreover, our statistical analyses for Figure 1C were within the brain region between genotypes and were never an intra-regional analysis. Therefore, these analyses were performed with relatively matching number of samples (11 vs. 8 in GP, 4 vs. 4 in SN and cerebellum). We also included all the exact n numbers in the figure legends to clarify this issue.

3) I do not see the information that total protein yield per wet tissue weight was the same for WT and KO mouse GP included in the manuscript/SM. This is useful and should be added (perhaps as a statement in the methods?)

We have added this information in the Materials and Methods section.

4) In regards to the localization of PPCS/PPCDC, the authors state in the rebuttal: "We performed new experiments and evaluated public databases to support our conclusion that PPCS is non-exclusively found in mitochondria and PPCDC is in cytoplasm. Moreover, we analyzed their subcellular localization in each brain region and found no differences nor any significant regional variation in their quantities."

However, apart from the following statement found in the discussion:

"The non-exclusive mitochondrial localization of several CoA synthesis enzymes, including PANK2, PPCS, and COASY, raises the intriguing possibility that CoA pathway intermediates such as 4'-phosphopantetheine might traverse organellar membranes and serve to replenish compartmental pools of CoA"

I do not see anything in the revised manuscript that reflects the outcome of these "new experiments" and analysis. The authors do provide references associated with this statement, but since they have gone through the trouble of doing the analysis I want to suggest that they are explicit with the reasons/basis for the statement.

We now also reference our own studies.

In addition, I find the use of "non-exclusive mitochondrial localization" in the quoted statement somewhat problematic. First, this phrase also refers to PANK2, which is stated to be UNIQUELY localized in the mitochondria in the introduction. Using "uniquely" in the first instance and then "nonexclusively" here seems contradictory; if it isn't, the authors should be clear in what they mean so that there is no confusion.

Second, either PPCS or PPCDC *must* be exclusively localized in one of the subcellular compartments in order to give credence to the idea that there are compartmental pools of CoA. Do the authors propose that PPCDC is not found in the mitochondria? If so, they again must be explicit in this regard, and explain their reasoning for this proposal.

We have clarified the language in both places in the manuscript.

5) Gene/Protein nomenclature: The possible confusion here is in the reference of mouse vs human genes/proteins. On p.18, in the line "With higher doses of 4'-phosphopantetheine, we observed progressively lower levels of Coasy expression in GP" I believe Coasy should be in italics.

We thank the Reviewer for directing us to this error in formatting, which now has been corrected.

6) "phosphopantothenoyleysteine" is spelled without a hyphen.
We have corrected this in the text.

We include updated data for Figure 4B. We removed the restriction in E-14 on Author's checklist because we expanded our fresh blood COASY QRT-PCR analysis from 'classical PKAN only' to 'all PKAN samples'. This has changed our Figure 4B and strengthened our statistical analysis. We have updated Author's checklist, main text, figure, and legend accordingly.

	n	n
	Previously	Updated
Control	11	51
PKAN	9	35

	Previously	Updated
P	0.00595077	0.0000000282

Corresponding Author Name: Hayflick
 Journal Submitted to: EMBO Molecular Medicine
 Manuscript Number: EMM-2019-10489